# Dynamic Agent Skills: A Lifecycle Survey and Taxonomy of Evolving Skill Libraries

**Yubo Li**
*Carnegie Mellon University*
*yubol@andrew.cmu.edu*

**Reviewed on OpenReview:** *https://openreview.net/forum?id=cjU3YbcRr8*

## Abstract

Large language model agents increasingly store reusable procedures outside the model. These reusable procedures are often called *skills*: they may be code functions, natural-language instructions, SKILL.md packages, workflow graphs, or learned adapters that a future agent can retrieve and invoke. This taxonomy-driven survey asks how such skill libraries change over time. Across a 124-paper 2023–2026 audit set, we synthesize dynamic skill systems as *lifecycle-managed, verified, evolving artifact stores*: agents collect evidence from interaction, propose skill updates, verify and admit candidates, organize them for retrieval and composition, repair or prune stale entries, and govern sharing through provenance and rollback. We organize the literature around three survey tools. First, a six-sense taxonomy distinguishes the structurally different artifacts called "skills" in current papers. Second, an eight-stage lifecycle architecture identifies the recurring design decisions behind evidence acquisition, proposal, verification/admission, storage, retrieval/composition, maintenance, distillation/portability, and governance. Third, a lightweight skill-record schema and ten-operator vocabulary provide common terms for comparing library updates without elevating them into a separate method contribution. Using this structure, we synthesize evidence-graded patterns with explicit caveats: admission and repair are repeatedly important, verifier quality materially affects skill-aware RL, flat retrieval can degrade as libraries grow, and current benchmarks still under-report library trajectories, usage–utility gaps, and safety surfaces. We close with concrete reporting standards and open problems for evaluating dynamic skills as changing libraries rather than static prompt or tool collections.

## 1 Introduction

Large language model agents are moving from chat-style assistance into operational workflows. They browse and manipulate web interfaces, write and repair software, operate desktop and computer-use environments, coordinate multi-step research pipelines, interact with embodied environments, and assist domain-specific workflows such as recommendation and medical imaging. Across these settings, the bottleneck is not only whether the model can reason about a single task. It is whether the agent can reuse procedural knowledge across recurring tasks, tools, interfaces, and users instead of re-deriving the same strategy inside every context window.

*Agent skills* were proposed as a practical answer to this reuse problem. In plain terms, a skill is a reusable procedure that an agent can call later. It may be a function, a natural-language instruction, a SKILL.md directory, a workflow graph, or a learned adapter, but its role is the same: preserve a reusable way of acting so that a future agent can retrieve, compose, and execute it. Early executable libraries such as VOYAGER and LATM stored code skills; 2025 systems extended skills into web and software agents; 2026 systems expanded the ecosystem into SKILL.md packages, computer-use skills, mobile GUI skills, multimodal skills, benchmarks, registries, and safety audits (Wang et al., 2023; Cai et al., 2023; Zheng et al., 2025a; Xia et al., 2025a; Chen et al., 2026d;b; Jiang et al., 2026a; Xie et al., 2026; Tao et al., 2026; Fan et al., 2026).

The broader skill ecosystem now includes SKILL.md conventions, function-calling schemas, MCP/plugin-style packaging, and registry-like platforms that support libraries with hundreds or thousands of reusable artifacts (Jiang et al., 2026b; Zhang et al., 2025; Anthropic, 2025; Li et al., 2026c; Zheng et al., 2026).

The word "skill" is overloaded. Table 1 is therefore the conceptual entry point for the survey: it distinguishes executable programs, natural-language lessons, SKILL.md packages, parametric adapters, memory traces, and capability labels before the paper introduces lifecycle or update notation. The first four senses form the main dynamic-skill cluster studied in this survey; the latter two are boundary cases included when they illuminate library behavior.

| Method | Sense of "skill" | Artifact form | Exec. | Edit | Portable | Inspect. | Verif. handle |
|---|---|---|---|---|---|---|---|
| VOYAGER (Wang et al., 2023) | Executable code | .py library | ✓ | ✓ | ∼[a] | ✓ | unit test |
| LATM (Cai et al., 2023) | Executable code | Python tool | ✓ | ✓ | ∼[a] | ✓ | unit test |
| AGENTFACTORY (Zhang et al., 2026i) | Executable code | Python + MCP | ✓ | ✓ | ∼[a] | ✓ | meta-agent inspect. |
| LIVE-SWE (Xia et al., 2025a) | Executable code | code snippet | ✓ | ✓ | ∼[a] | ✓ | rollout |
| SAGE (Wang et al., 2025a) | Executable code | Python function | ✓ | ✓ | ∼[a] | ✓ | env. reward |
| ASI (Wang et al., 2025b) | Executable code | program skill | ✓ | ✓ | ∼[a] | ✓ | rollout |
| HASP (Liu et al., 2026b) | Executable code | program function | ✓ | ✓ | ∼[a] | ✓ | teacher/rollout |
| SKILLOPS (Song et al., 2026b) | Executable contract | typed skill contract | ✓ | ✓ | ∼[a] | ✓ | validator/graph audit |
| ERL (Allard et al., 2026) | NL heuristic | critique | − | ✓ | ✓ | ✓ | indirect |
| RETROAGENT (Zhang et al., 2026g) | NL heuristic | dual lesson | − | ✓ | ✓ | ✓ | indirect |
| EVOLVER (Wu et al., 2025) | NL heuristic | strategic principle | − | ✓ | ✓ | ✓ | indirect |
| METACLAW (Xia et al., 2026b) | SKILL.md | L1/L2/L3 markdown | ✓[b] | ✓ | ✓ | ✓ | L3 code gate |
| MEMENTO (Zhou et al., 2026a) | SKILL.md + case | markdown + trace | ✓[b] | ✓ | ✓ | ✓ | L3 code gate |
| TRACE2SKILL (Ni et al., 2026) | SKILL.md | L2 markdown | ✓[b] | ✓ | ✓ | ✓ | rubric judge |
| AUTOREFINE (Qiu et al., 2026) | SKILL.md | dual-form markdown | ✓[b] | ✓ | ✓ | ✓ | rubric judge |
| SKILLFLOW-BENCH (Zhang et al., 2026j) | SKILL.md patch | files + JSON patch | ✓[b] | ✓ | ✓ | ✓ | benchmark verifier |
| SKILLOPT (Yang et al., 2026b) | SKILL.md / text skill | optimized document | − | ✓ | ✓ | ✓ | held-out utility |
| SKILLGRAD (Wang et al., 2026b) | SKILL.md | structured package | ✓[b] | ✓ | ✓ | ✓ | loss/validation |
| MUSE-AUTOSKILL (Lin et al., 2026) | SKILL.md + memory | skill + per-skill memory | ✓[b] | ✓ | ✓ | ✓ | unit tests |
| CONTRACTSKILL (Lu et al., 2026b) | Contract skill | pre/postcondition artifact | ✓ | ✓ | ✓ | ✓ | deterministic checks |
| ABSTRAL (Song et al., 2026a) | SKILL.md / design doc | structured markdown | − | ✓ | ✓ | ✓ | trace evidence |
| SKILLSCRAFTER (Wang et al., 2026f) | Parametric | LoRA subspace | ✓[c] | − | − | − | behavioral probe |
| SKILL0 (Lu et al., 2026a) | Parametric | adapter | ✓[c] | − | − | − | behavioral probe |
| SELAUR (Zhang et al., 2026b) | Parametric | LoRA | ✓[c] | − | − | − | behavioral probe |
| LSE (Chen et al., 2026e) | Parametric (mix) | prompt + weight | ✓[c] | ∼ | − | ∼ | behavioral probe |
| SIMPLEMEM (Liu et al., 2026d) | Memory / trajectory | trace case | − | ∼ | ✓ | ✓ | indirect |
| MUSE (Yang et al., 2025a) | Memory / trajectory | episodic store | − | ∼ | ✓ | ✓ | indirect |
| CASCADE (Huang et al., 2025) | Memory / trajectory | curated case set | − | ∼ | ✓ | ✓ | indirect |
| XSKILL (Jiang et al., 2026a) | Skill + experience | MD skill + JSON exp. | ∼ | ✓ | ✓ | ✓ | indirect |
| SKILLFLOW-2025 (Li et al., 2025) | Registry-retrieved skill | SKILL.md corpus + ranked candidates | ∼ | − | ✓ | ✓ | retrieval eval |
| SRA (Su et al., 2026) | Registry-retrieved skill | large skill corpus | ∼ | − | ✓ | ✓ | retrieval + utility eval |
| SKILLSVOTE (Liu et al., 2026c) | Registry-governed skill | executable + guidance corpus | ∼ | ✓ | ✓ | ✓ | evidence gate |
| SSL (Liang et al., 2026a) | Structured representation | JSON-like skill graph | − | ✓ | ✓ | ✓ | support audit |
| CO-EVOLVING (Jung et al., 2025) | Capability label | declarative role | − | ∼ | ✓ | ✓ | none |
| GEA (Weng et al., 2026) | Capability label | declarative role | − | ∼ | ✓ | ✓ | none |

Table 1: **Representative dynamic-skill methods keyed by paper, with the sense of "skill" each adopts and the five properties that determine its dynamic behaviour.** The six senses partition the design space (horizontal rules) and project directly onto the taxonomy axes of Section 6; the first four senses are the "dynamic skills" studied in this survey while the last two are boundary cases with restricted edit / verification dynamics. Column definitions: *Exec.* = is the artifact machine-executable; *Edit* = can be revised without retraining; *Portable* = usable across different LLM backbones; *Inspect.* = auditable in textual form; *Verif. handle* = form of admission check available. Symbols: ✓ = fully satisfies; ∼ = partial / indirect; − = does not satisfy. [a]portable across backbones that can call the same runtime; [b]executable only when L3 attaches code or MCP resources; [c]executable only through the compatible base model.

This adoption changes the research question. In OpenClaw-style personal agents,[1] skills are no longer helpful prompt snippets; they are part of the execution substrate. SKILLCLAW uses cross-user OpenClaw interactions to evolve shared skills over deployment rounds (Ma et al., 2026b), while CLAWSAFETY shows that skill files can become a high-trust prompt-injection channel in privileged personal-agent scaffolds (Wei et al., 2026). The same abstraction that improves reuse creates questions of verification, maintenance, provenance, and governance.

---

[1]We use OpenClaw, SKILLCLAW, and CLAWSAFETY as research-artifact or evaluation-setting names from the cited papers; the survey relies on the papers' methods and measurements, not on product or platform claims.

Many skill libraries are still treated as *static*: authored once, versioned rarely, and weakly connected to the trajectories that reveal whether a skill remains useful, safe, or correct. Recent work (Alzubi et al., 2026; Zheng et al., 2025a; Wang et al., 2025b; 2026g; Lu et al., 2026b; Chen et al., 2026b; Wang et al., 2025a; Ni et al., 2026; Shi et al., 2026; Li et al., 2026b; Song et al., 2026b; Li et al., 2025; Xia et al., 2026b; Yang et al., 2026b; Wang et al., 2026b; Lin et al., 2026) rejects this assumption by treating the library itself as a learning object that grows, shrinks, repairs itself, changes storage structure, and sometimes distills its contents back into model weights. The methods differ—retrospective consolidation, execution-grounded honing, RL-based proposal, rollback-validated refactoring, registry-scale retrieval, structured repair, library-time technical-debt maintenance, text-space skill optimization, and two-timescale distillation—but share the commitment that skill libraries should be managed as evolving systems.

This survey synthesizes the field through a lifecycle view: dynamic skill systems are *lifecycle-managed, verified, evolving artifact stores for LLM agents.* A skill artifact is created from evidence, proposed as an edit, verified, admitted, organized, retrieved or composed, maintained, sometimes distilled, and governed through provenance and rollback. Papers differ mainly in which stages they implement, which learning signal drives edits, and where they place the verifier.

The result is a taxonomy-driven survey rather than an encyclopedic paper-by-paper catalog: the goal is to make artifact boundaries, lifecycle commitments, update mechanisms, and evidence strength comparable across a fast-moving literature.

## 1.1 Dynamic skill libraries as learning systems

A *library* is a collection of such skills equipped with a retrieval or composition mechanism. A *dynamic* library is one whose contents or organization can change as the agent accumulates evidence. We use a compact record schema and operator vocabulary later in the paper to make this comparison precise, but the basic idea is simple: a dynamic skill system does not only choose which skill to read at inference time; it changes the store that future agents will read from.

This makes dynamic skills a learning-systems problem rather than merely a software-packaging problem. Interaction produces evidence; evidence selects edits; edits change the library; the changed library alters the distribution of future actions through retrieval, composition, and execution. In this view, $\mathcal{L}_t$ is part of the agent's state, the transition $\mathcal{L}_t \rightarrow \mathcal{L}_{t+1}$ is a learning rule, verification is a selection mechanism, maintenance is a form of regularization, and distillation is a consolidation step that moves external procedural knowledge into slower parametric or semi-parametric stores. The same library can also serve as the interface between individual learning and collective learning when skills are shared across users, registries, or agents.

The time index matters because deployed libraries go stale as tasks, tools, backbones, and safety constraints drift. It also makes the evidence-graded patterns in §9 expressible: admission and verifier quality, retrieval degradation, weaker-backbone gains, focused curation, maintenance at scale, and write-time abstraction are all properties of a changing library.

## 1.2 Why a survey now

Three developments in 2025–2026 made dynamic skills surveyable. First, packaging and registry infrastructure matured enough that libraries must be curated, routed, and governed as collections (§10.3). Second, methods now span the learning-signal spectrum, from LLM-as-judge admission through audited verification and two-timescale RL. Third, deployment and benchmark papers report failure modes—retrieval degradation, skill injection, verification drift, skill inflation, and maintenance-off collapse—that earlier method papers treated mostly in ablation. SKILLFLOW-BENCH (Zhang et al., 2026j), for example, evaluates discovery, patching, reuse, and maintenance over sequential tasks and shows that high skill-use rates need not imply high utility.

The field still lacks shared vocabulary and benchmark protocol. Papers use "skill" for at least six structurally different artifacts; update operations are named inconsistently; benchmarks often report terminal performance rather than library trajectories (§8). The survey's role is to provide a common language before ranking is meaningful.

This paper is complementary to prior surveys. Xu & Yan (2026) provide a broad account of agent skills across architecture, acquisition, security, and future directions; Fang et al. (2025) survey self-evolving agents beyond skill libraries; Zheng et al. (2025c) emphasize lifelong LLM-agent learning with a memory-centric lens; and Zhou et al. (2026b) provide a broad May 2026 taxonomy of agent-skill techniques and applications. Our focus is narrower: dynamic, lifecycle-managed skill libraries and the mechanisms by which such libraries change over time.

### 1.3 Contributions

This paper makes five contributions. *First*, it clarifies what current papers mean by "skill" through the six-sense taxonomy in Table 1. *Second*, it recasts dynamic skills as editable learning-system artifacts and introduces a lightweight skill-record schema plus library-level transition notation for comparing systems (§3). *Third*, it introduces a lifecycle architecture spanning evidence acquisition, proposal, verification/admission, organization, retrieval/composition, maintenance/repair, distillation/portability, and governance (§5). *Fourth*, it uses a ten-operator vocabulary {ADD, REFINE, MERGE, SPLIT, PRUNE, DISTILL, ABSTRACT, COMPOSE, REWRITE, RERANK} to organize mechanisms and system families without claiming closure or composition laws (§§6–7). *Fifth*, it audits lifecycle-aware evaluation, then distills seven evidence-graded patterns, eight safety surfaces, and eight open problems tied to concrete next experiments (§§8, 9, 11, 12).

## 2 Corpus, Scope, and Review Protocol

We separate the object of study from the process used to assemble it. The survey covers papers that treat skill *libraries* or skill-like external artifacts as dynamic objects: artifacts may be added, revised, merged, pruned, routed, distilled, transferred, or governed as the agent accumulates evidence. It excludes classical option discovery, generic tool-use benchmarking, and fine-tuning pipelines unless they produce an externally invocable skill artifact or directly evaluate such artifacts' lifecycle.

### 2.1 Inclusion Criteria

The working audit set contains 124 modern papers after the May 31, 2026 update: 2 from 2023, 19 from 2025, and 103 from 2026. This set includes the primary dynamic-skill method, benchmark, infrastructure, and safety cluster, plus boundary/context papers that define terminology, adjacent lifelong-agent evaluation, or neighboring self-evolution settings. Older classical-options and hierarchical-RL anchors are cited for comparison but are not counted in this modern audit set. We included papers from 2023–2026 when they satisfied at least one of three criteria: (i) the paper proposes a mechanism that adds, edits, prunes, distills, routes, transfers, or composes an agent skill artifact; (ii) the paper provides infrastructure or a benchmark that changes how skill libraries are stored, retrieved, evaluated, or governed; or (iii) the paper documents a safety or deployment failure mode specific to skill artifacts. Boundary/context works — the systematization-of-knowledge (SoK) paper on agentic skills (Jiang et al., 2026b), Xu & Yan (2026)'s 2026 position paper, MAGELLAN's autotelic curriculum mechanism (Gaven et al., 2025), the ELL/STULIFE benchmark (Cai et al., 2025), and the EXPERIENCE COMPRESSION SPECTRUM framework (Zhang et al., 2026h) — are retained as scope-setting evidence rather than as members of the primary method cluster.

Figure 2 visualizes the corpus at the level needed for survey synthesis: a sparse 2023 foundation, no primary 2024 item after filtering, a 2025 transition wave, and a dense 2026 expansion into executable libraries, infrastructure, benchmarks, and safety.

### 2.2 Search and Screening Procedure

The corpus was assembled by iterative database search and backward/forward snowballing rather than by a single PRISMA-style query. We searched arXiv, Semantic Scholar, Google Scholar, OpenReview, and venue proceedings using combinations of *LLM agent skill, agentic skill, skill library, self-evolving agent, lifelong agent, skill routing, SKILL.md, agent skill safety, skill benchmark*, and named-system queries for

papers discovered during snowballing. Candidate papers were screened first by title/abstract and then by PDF when the abstract suggested a persistent artifact, skill-like package, evolving library, or skill-specific safety/evaluation surface. We excluded papers whose only adaptation object was model weights, prompt optimization, generic tool use, or episodic memory without an invocable interface, unless the paper directly evaluated how such artifacts become reusable skills. Because the collection was assembled in an arXiv-heavy and still-moving area, we report the final audit set and inclusion frontier rather than a PRISMA exclusion flow. The survey does not claim an exhaustive count of every tool-use, memory, or HRL paper adjacent to the topic.

## 2.3 Temporal Cutoff and Update Policy

All statements in the main text should be read against an explicit temporal cutoff: May 31, 2026. Papers appearing after that date are outside the surveyed corpus. The cutoff is an audit boundary, not a signal that the field has stabilized. We distinguish *corpus updates*, which add papers satisfying the criteria above, from *synthesis updates*, which revise the taxonomy, evidence-graded patterns, or open problems only when new evidence changes a lifecycle stage or exposes a new failure mode. Product and specification sources such as the public Agent Skills format are cited only for packaging facts and are not counted as papers in the audit set.

## 2.4 Note Schema and Conflict Resolution

Each included paper was read into a common note schema covering problem framing, mechanism, control action, experimental setup, headline results, ablations, limitations, closest peers, and survey takeaway. When duplicate notes disagreed, we resolved the conflict against the PDF. The schema is important because several names are close but technically distinct: SKILLFLOW-2025 is a multi-stage retrieval system for community skill repositories, while SKILLFLOW-BENCH is a lifelong skill-discovery benchmark; SAGE stores executable Python skills, while PSN's refactors are rollback-validated graph edits.

## 2.5 Evidence Roles

The corpus mixes method papers, infrastructure papers, benchmarks, and safety studies. We use method papers for evidence about operators, triggers, verifiers, and mechanisms; benchmark papers for evaluation protocol and lifecycle failure modes; infrastructure papers for storage, portability, registries, and operator cost; and safety papers for attack surfaces and partial defenses. We grade evidence qualitatively: **A** means multiple controlled ablations or one clean ablation plus independent corroboration; **B** means one controlled study or a strong benchmark/deployment measurement; **C** means convergent benchmark behavior without a clean causal ablation; and **D** means architectural corroboration only. We avoid cross-paper leaderboards unless the harness is shared, because backbone, tool surface, context budget, and evaluator often change together.

## 2.6 Relation to Adjacent Surveys

Four peer surveys cover adjacent territory and we do not duplicate their full scope. Xu & Yan (2026) surveys agent-skill architecture, acquisition, and security from a broad position-paper vantage; Fang et al. (2025) surveys self-evolving agents beyond skill libraries; Zheng et al. (2025c) surveys lifelong learning for LLM agents with a memory-centric emphasis; and Zhou et al. (2026b) provides a broad May 2026 taxonomy of agent-skill techniques and applications. Our contribution relative to these is the library-as-object framing, the lifecycle architecture, and the operator-level vocabulary: instead of asking whether agents can learn over time in general, we ask how an externally invocable artifact store changes, verifies, maintains, and governs itself.

The scope also separates this survey from classical options and hierarchical reinforcement learning. Options, skill chaining, option-critic methods, and deep skill-discovery methods study temporally extended policies inside an environment (Sutton et al., 1999; Konidaris & Barto, 2009; Bacon et al., 2017; Eysenbach et al., 2019; Nachum et al., 2018). Dynamic agent skills add a different object: an externally inspectable artifact

that can be edited after deployment, packaged with metadata, admitted or rejected by language- or execution-level verifiers, shared across agents, and governed through provenance. We use the options framework as a starting point because it clarifies applicability, policy, and termination; the survey's added components are the artifact-store machinery that classical HRL usually leaves implicit.

## 3 Skill Records and Library Updates

The preceding section and Table 1 establish the terminology. This section gives the lightweight notation used in the rest of the survey. The notation is a comparison scaffold, not a separate model of skill learning: it lets us say which artifact is being edited, which verifier admits it, which lineage record survives, and how the library changes from one state to the next.

### 3.1 The six senses of "skill"

Table 1 partitions the literature along five dynamic properties: whether the artifact is *executable*, *editable in place*, *portable across models*, *inspectable by humans*, and attached to a *verification handle*. These properties largely determine which triggers, operators, and signals a method can support.

The executable-code sense, occupied by Voyager, LATM, LIVE-SWE, AgentFactory, SAGE, ASI, HASP, and SkillOps (Wang et al., 2023; Cai et al., 2023; Xia et al., 2025a; Zhang et al., 2026i; Wang et al., 2025a;b; Liu et al., 2026b; Song et al., 2026b), treats a skill as a named program or typed contract that can be invoked, inspected, edited line-by-line, and verified by tests, validators, or environment feedback. The NL-heuristic sense, occupied by ERL, RetroAgent, EvolveR, and EmbodiSkill (Allard et al., 2026; Zhang et al., 2026g; Wu et al., 2025; Ju et al., 2026), treats a skill as a short lesson or procedural specification injected at retrieval time; it is editable and portable but only indirectly verifiable through downstream task success. The SKILL.md sense, occupied by MetaClaw, ABSTRAL, Trace2Skill, Memento, K2-Agent, AutoRefine, SkillFlow-Bench, SkillOpt, SkillGrad, and MUSE-Autoskill (Xia et al., 2026b; Song et al., 2026a; Ni et al., 2026; Zhou et al., 2026a; Wu et al., 2026b; Qiu et al., 2026; Zhang et al., 2026j; Yang et al., 2026b; Wang et al., 2026b; Lin et al., 2026), unifies code and prose behind a progressive-disclosure package: L1 metadata for routing, L2 instructions for the agent, and L3 code, schemas, or sub-skills for execution and verification. Product/specification sources describe the public packaging convention as a directory with a `SKILL.md` descriptor, optional scripts, and progressive disclosure from metadata to full instructions and resources (Zhang et al., 2025; Anthropic, 2025); we use those sources only for format facts, not as evidence for lifecycle effectiveness.

The parametric sense, represented by SkillsCrafter, SELAUR, SKILL0, and LSE (Wang et al., 2026f; Zhang et al., 2026b; Lu et al., 2026a; Chen et al., 2026e), treats a skill as a weight delta, trained prompt module, or internalized procedure; such artifacts are probed behaviorally, ported only across compatible base models, and edited through training. The memory/trajectory sense, present in SimpleMem, MUSE, CASCADE, XSkill, SkillTTA, and the case layer of Memento (Liu et al., 2026d; Yang et al., 2025a; Huang et al., 2025; Jiang et al., 2026a; Wang et al., 2026d; Zhou et al., 2026a), blurs skills with retrievable episodes; the boundary is whether the trace has an invocable interface. The registry-retrieved and capability-label senses, present in SkillFlow-2025, SkillsVote, SkillsInjector, Co-Evolving-Agents, and some of GEA (Li et al., 2025; Liu et al., 2026c; Li et al., 2026d; Jung et al., 2025; Weng et al., 2026), are the most brittle under dynamic evolution because retrieval success or claimed capability has no robust edit body unless paired with executable skill contents and admission checks.

In the rest of the paper, "skill" means executable code, NL heuristic, SKILL.md package, or parametric skill by default; memory/trajectory and capability-label artifacts are treated as boundary cases.

This definition also separates *skills* from ordinary *tools*. A tool is usually an externally supplied callable resource: the agent decides when to invoke it, but the tool's body, interface, and release process are not learned by the agent. A dynamic skill is a persistent artifact whose lifecycle is itself part of learning. The same Python function or MCP endpoint can therefore be a tool in a generic tool-use benchmark and a skill in a dynamic-skill system if the agent proposes it, revises it from trajectories, verifies it for admission,

stores lineage, and later maintains or transfers it. The distinction is operational rather than syntactic: what matters is whether the artifact participates in $\mathcal{L}_t \to \mathcal{L}_{t+1}$.

## 3.2 From static skill records to editable records

**Starting point.** The canonical reference for skills as temporally extended actions is the options framework of Sutton et al. (1999): an option is a triple $\langle I, \pi, \beta \rangle$ of an initiation set $I \subseteq \mathcal{S}$, a policy $\pi$, and a termination condition $\beta$. Within the LLM-agent literature, the systematization-of-knowledge paper by Jiang et al. (2026b) reinterprets this triple for language agents as a four-tuple

$$\mathcal{S} = \langle C, \pi, T, R \rangle, \tag{1}$$

where $C$ is an applicability predicate ("when is this skill relevant"), $\pi$ is the executable policy (code, prompt template, adapter, or lesson text), $T$ is a termination condition (success, exception, or budget exhaustion), and $R$ is a *reusable interface*: the invocation name, expected inputs, outputs, preconditions, return values, and composition points that let another agent, tool, or skill call the artifact.

**Five concrete gaps.** The SoK-Skills four-tuple is adequate for *static* libraries, but dynamic libraries require five missing objects. First, a *time index* distinguishes $\mathcal{S}_t$ from later refinements, replacements, and merges. Second, an *edit operator* records whether a method rewrites NL lessons (ERL, EvolveR, RetroAgent), function bodies (SAGE), SKILL.md prose (AutoRefine, Memento), symbolic refactors (PSN) (Shi et al., 2026), or mutation heuristics (CODE-SHARP) (Bornemann et al., 2026). Third, an *admission gate* captures filters such as EvoSkill's Pareto front (Alzubi et al., 2026), SkillCraft's MCP verifier (Chen et al., 2026c), and ASG-SI's audited graph (Huang & Huang, 2025). Fourth, *lineage* supports rollback, supersession, and maturity gating in systems such as AgentDevel, PSN, Memento, and Trace2Skill (Zhang, 2026; Shi et al., 2026; Zhou et al., 2026a; Ni et al., 2026). Fifth, a *library-level object* is needed because the main transition is $\mathcal{L}_t \to \mathcal{L}_{t+1}$, not merely one tuple to another.

**The extended tuple.** We therefore represent an editable skill record with seven fields

$$\mathcal{S}_t = \langle C_t, \pi_t, T_t, R_t, \varphi_t, \nu_t, \prec_t \rangle, \tag{2}$$

with the new components interpreted as skill-record fields. The tuple should not be read to mean that every skill owns an independent learning algorithm. In most systems, the update policy is library-level, while individual skill records expose the handles that policy can use: editable body, verification evidence, interface, and lineage.

The *edit field* $\varphi_t$ records how a candidate revision can be generated for this artifact under an edit instruction $u_t$ (chosen from the operator vocabulary introduced below in §3.3). In deterministic systems, $\varphi_t : \mathcal{S}_t \times u_t \to \mathcal{S}_{t+1}$ maps directly to a successor skill. In stochastic proposal systems such as mutation-based search, $\varphi_t$ is better read as a proposal kernel $q_t(\mathcal{S}' \mid \mathcal{S}_t, u_t)$, from which a realized candidate $\tilde{\mathcal{S}}_{t+1}$ is sampled before admission. The edit component therefore need not guarantee improvement; the admission predicate below decides whether the realized candidate changes library state. The *verification field* $\nu_t : \mathcal{S}_t \to \{0, 1\}$ records the available admissibility handle—whether a proposed or edited skill can pass a quality gate before entering $\mathcal{L}_{t+1}$—and may be a unit test (LATM), a grounded rollout (SkillWeaver, EvoSkill), a meta-agent inspection (AgentFactory), an ensemble judge (Trace2Skill, AgentSkillOS), a static analyzer (SkillCraft), a symbolic audit (ASG-SI), or a Bayesian prior over future utility (CODE-SHARP). The *lineage relation* $\prec_t$ records which version supersedes which; methods that support rollback, A/B maturity, or blast-radius limits require an explicit $\prec$.

A static library is the limiting case in which no update trigger admits a library edit, so $\mathcal{L}_{t+1} = \mathcal{L}_t$ except for exogenous releases. An unconditional verifier with repeated Add is not static; it is an unfiltered append-only store. The artifact type determines feasible $(\varphi, \nu, \prec)$ choices: executable skills support machine-checkable gates, NL heuristics require indirect verification, parametric skills move edits onto a training timescale, and capability labels usually lack a well-defined edit operator. Cross-skill operations such as composition, merging, and splitting are library-level transitions, so we describe them next.

### 3.3 Library dynamics: $\mathcal{L}_t \to \mathcal{L}_{t+1}$

A dynamic skill system is a library plus rules for how that library changes. Write

$$\mathcal{L}_t \;=\; \{\mathcal{S}_t^{(1)}, \mathcal{S}_t^{(2)}, \ldots, \mathcal{S}_t^{(N_t)}\} \cup \mathcal{M}_t, \tag{3}$$

where $\mathcal{M}_t$ holds auxiliary metadata: invocation statistics, call graphs, maturity labels, verifier caches, and any cross-skill edges (prerequisite, composition, shared resources, author). The library transition is then

$$\mathcal{L}_{t+1} \;=\; \mathcal{T}(\mathcal{L}_t, \, \tau_t, \, r_t) \;=\; \mathrm{Apply}(\vec{u}_t(\tau_t, r_t), \, \mathcal{L}_t), \tag{4}$$

where $\tau_t$ is an *evolution trigger* (a timer, a task boundary, a failure event, a user edit), $r_t$ is the *learning signal* the system uses to choose an edit (task reward, natural-language critique, self-judgment, cross-user aggregate, teacher signal), and $\vec{u}_t$ is a vector of operator-instruction pairs drawn from the ten-element vocabulary

$$\vec{u}_t \subseteq \{\text{ADD}, \text{REFINE}, \text{MERGE}, \text{SPLIT}, \text{PRUNE}, \text{DISTILL}, \text{ABSTRACT},$$
$$\text{COMPOSE}, \text{REWRITE}, \text{RERANK}\} \times \text{Instruction}.$$

The ten operators have fixed meanings throughout the paper: ADD inserts a skill; REFINE edits content without changing the interface; MERGE combines skills; SPLIT factors one skill into components; PRUNE removes or quarantines; DISTILL compresses trajectories into a skill; ABSTRACT lifts a concrete procedure to a template; COMPOSE chains skills into a composite; REWRITE changes the body and possibly the interface; and RERANK changes retrieval priors without changing content. Representative instances include VOYAGER and SAGE for ADD, MEMENTO, PSN, SKILLOPT, and SKILLGRAD for REFINE, TRACE2SKILL and AUTOREFINE for MERGE, SKILLX for SPLIT, WILD-SKILLS and CLAWSAFETY (Wei et al., 2026) for PRUNE, CASCADE and MUSE for DISTILL, CUA-SKILL, COEVOSKILLS, and SKILLGEN for ABSTRACT, SKILLCRAFT, SKILLORCHESTRA, and HASP for COMPOSE, EVOLVER and EMBODISKILL for REWRITE, and SKILLROUTER and SKILLSINJECTOR for RERANK. Almost no method implements all ten; the supported subset is one of the clearest taxonomic fingerprints.

**Verification as library-level gate.** Although $\nu$ is indexed per skill record in Equation 2, verification acts at *admission* time: an edit yields a candidate $\mathcal{S}^*$, and $\mathcal{S}^*$ enters $\mathcal{L}_{t+1}$ only if the library-level admission policy accepts the candidate using the available verification handle. This edit-versus-admission distinction underlies the verification architecture of Section 7 and the R1 pattern in Section 9.

**Two timescales.** The trigger $\tau_t$ can be a per-step failure (SELAUR), per-task retrospective pass (SAGE, ERL), periodic maintenance cycle (AUTOREFINE), or release decision (AGENTDEVEL). Parametric systems such as METACLAW, SKILL0, and LSE add a slow loop in which many fast-loop library updates are distilled into weights or adapters.

**Scope.** Throughout the rest of the paper, "dynamic skill" refers to a skill $\mathcal{S}_t$ with an editable body, an admission/verification handle, or lineage metadata, or to a library $\mathcal{L}_t$ that evolves under a non-empty update vector $\vec{u}_t$. A *static* library is the special case $\vec{u}_t \equiv \varnothing$. The survey addresses the dynamic case; static-library methods are included only as baselines or as infrastructure substrates on which dynamic methods are built.

### 3.4 Worked instantiation: AutoRefine as a library transition

The notation is intended to describe concrete system behavior, not just to name components. Consider an AUTOREFINE-style maintenance cycle (Qiu et al., 2026). A skill document in the current library can be written as

$$\mathcal{S}_t^{(i)} = \langle C_t, \pi_t, T_t, R_t, \varphi_t, \nu_t, \prec_t \rangle,$$

where $C_t$ is the task or state description under which the skill should be retrieved, $\pi_t$ is the SKILL.md instruction body plus optional executable helper, $T_t$ is the success/failure or budget condition observed during use, and $R_t$ is the call signature or natural-language invocation handle. After a trajectory exposes a

failure or redundancy, the trigger is $\tau_t = \text{TASKEND}$ or $\text{PERIODIC}$, and the signal $r_t$ is a critique, execution trace, or downstream utility measurement. The update rule chooses an operator vector such as

$$\vec{u}_t = \{(\text{REFINE}, \text{patch ambiguous step}),$$
$$(\text{MERGE}, \text{combine duplicate routines}),$$
$$(\text{PRUNE}, \text{quarantine low-utility skill})\}.$$

Each realized edit yields a candidate $\mathcal{S}^*$. The admission gate evaluates $\nu_t(\mathcal{S}^*)$ using the method's judge, execution feedback, or consistency checks. If the candidate passes, $\mathcal{S}^*$ enters $\mathcal{L}_{t+1}$ and $\prec_{t+1}$ records that it supersedes or merges earlier artifacts; if it fails, $\mathcal{L}_{t+1}$ retains the prior version or marks the candidate for later review. In this example, the transition in Equation 4 is not a single append operation: it is a gated composition of REFINE, MERGE, and PRUNE over a versioned artifact store.

## 4 Why Static Skill Libraries Fail

The record schema of §3 treats the skill library as a time-indexed object $\mathcal{L}_t$ because the static alternative fails in recurring, connected ways. Static libraries are useful when the task distribution, tool surface, verifier, and authoring assumptions remain stable. The surveyed literature shows that long-lived agent deployments rarely satisfy those conditions. The failure is not one defect but a lifecycle collapse: authoring, verification, retrieval, provenance, and adaptation are all forced into a one-time design decision.

The first pressure is economic. Static skills require up-front human authoring, yet deployment value is only observed after the skill has been used in context. Deployment-facing papers such as SKILLCLAW, AUTOSKILL, and AGENTSKILLOS describe useful SKILL.md authoring as labor-intensive, and SOK-SKILLS observes that library quality is bounded by the weakest authors. The issue is not simply that documentation is costly; it is that a static library pays the cost per skill before knowing which skills will matter. Dynamic systems shift part of that cost to write-time ABSTRACT and DISTILL, so trajectory evidence decides what should become reusable.

The second pressure is that correctness is not stationary. A static library freezes the author's implicit verifier at authoring time, but tasks, tools, base models, runtimes, and safety constraints drift. PSN's refactor detectors depend on recent transition history; AGENTSKILLOS's capability-tree audits assume a current tool surface; CODE-SHARP's execution verifier assumes a compatible runtime. Once those assumptions move, a skill can remain syntactically valid while becoming operationally wrong. Dynamic systems make $\nu$ re-runnable and allow the admission gate to remove, quarantine, or demote skills whose verifier has drifted.

The third pressure appears as the library grows: selection becomes harder than storage. Controlled size sweeps show that flat retrieval can degrade in the moderate-library-size regime, often around tens to hundreds of skills. SINGLE-AGENT-SKILLS gives the cleanest controlled curve, while WILD-SKILLS, SKILLROUTER, and AGENTSKILLOS show related degradation under realistic distractors, 80K-scale routing, and large flat stores (Li, 2026; Liu et al., 2026h; Zheng et al., 2026; Li et al., 2026b). The exact threshold varies, but the mechanism is stable: adding skills eventually adds distractors faster than utility. A static system can cap the library or redesign retrieval, but it has no native way to ask whether old skills should be merged, pruned, or reranked. Dynamic systems add those maintenance operators: PRUNE, MERGE, and RERANK.

The fourth pressure is provenance. Ordinary version control records who edited a file and when, but not which trajectory, verifier, or deployment signal justified admission. That gap matters when a skill regresses (AUTOREFINE, AGENTSKILLOS), when admission must be re-run against a new verifier, or when a cross-user skill must be attributed, redacted, or rolled back (SKILLCLAW, AUTOSKILL). PSN's rollback gate makes the point concrete: tentative refactors are reverted if success on three recent tasks drops by more than 20%. The lineage relation $\prec$ is the minimal structural addition that makes rollback, re-admission, and provenance tractable.

Finally, deployment itself is non-stationary. A static library is a snapshot of the authoring distribution, but real task mixes shift. The surveyed evidence adds two important shapes to this familiar observation: weaker backbones gain disproportionately from dynamic skills, and focused libraries become stale when the task

mix changes. The response is not merely to refresh the library occasionally; it is to specify an evolution trigger $\tau$ and a fast-loop clock that decide when evidence is allowed to alter the library.

These failures explain why the added record fields are not cosmetic. Authoring cost points to $\varphi$; verifier drift points to re-runnable $\nu$; retrieval pollution points to maintenance operators; attribution loss points to $\prec$; and task non-stationarity points to explicit triggers. Table 2 reads the same mapping from the architecture side, and Table 11 maps methods to the lifecycle gaps they address.

## 5 Dynamic Skill Systems as Lifecycle-Managed Stores

Section 4 gives the negative case: static skill libraries fail because they make authoring, verification, retrieval, provenance, and adaptation one-time decisions. The positive object is therefore not just a larger skill library. It is a controlled state machine over an evolving store of artifacts. A dynamic skill system observes interaction evidence, proposes a skill or library edit, verifies the candidate, admits it into a storage topology, retrieves or composes it at future invocation time, maintains it as the library ages, and records enough provenance to support rollback, transfer, and governance.

This section defines that reference architecture. The next section uses it as a taxonomy for the surveyed papers; Section 7 then analyzes the implementation choices that make particular lifecycle stages possible. Keeping these roles separate is important: the lifecycle is the *architecture*; the taxonomy is the *classification of systems*; the mechanism design space is the *choice of operators, verifiers, and clocks*.

### 5.1 Reference Architecture

Figure 1 gives the high-level architecture; Table 2 records the corresponding design questions, operators, evidence, and failure modes. We decompose a dynamic skill system into eight recurring stages. *Evidence acquisition* decides what observation can justify a library change: a trajectory, reward, failure trace, user edit, cross-user signal, or external resource. *Proposal* converts that evidence into a candidate artifact or edit. *Verification and admission* decides whether the candidate is allowed to enter the library, and whether it enters as mature, tentative, quarantined, or rejected. *Organization and storage* assigns the admitted artifact to a flat index, hierarchy, DAG, invocation graph, ontology, dual memory/skill store, or parametric subspace. *Retrieval and composition* decides which artifacts influence a future action. *Maintenance and repair* keeps the library compact and correct by pruning, merging, splitting, reranking, refining, or rewriting artifacts. *Distillation and portability* moves procedural knowledge between external artifacts, model weights, agents, users, or task domains. *Governance and provenance* records lineage, authorship, safety checks, release state, and rollback handles; in implementations this usually means audit logs over admitted candidates, verifier decisions, operator edits, and rollback events.

The stage order should not be read as a waterfall. Many systems loop between proposal and verification, perform retrieval before maintenance, or run distillation only periodically. The point is that each stage asks a different design question. A system that performs strong retrieval has not necessarily solved admission; a system that frequently uses a skill has not necessarily shown utility; a system that can add skills has not necessarily learned how to remove or repair them.

### 5.2 The Admission Boundary

The most important boundary in the lifecycle is between *candidate* artifacts and *admitted* library state. Dynamic systems can generate many plausible skills cheaply, but the library only improves when the admission gate filters them with an appropriate verifier. Execution-grounded systems such as SKILLWEAVER, EVOSKILL, and SKILLFOUNDRY place the gate close to write time; maintenance-heavy systems such as AUTOREFINE and AGENTSKILLOS re-run the gate as the library ages; rollback systems such as PSN treat admission as tentative until recent-task utility remains stable. This boundary explains why verification is not a side module. It is the selection mechanism that turns skill generation into library learning.

Admission also determines what provenance must be stored. If a skill was admitted because of a trajectory, validator, cross-user signal, or release audit, the system must retain that justification so that future mainte-

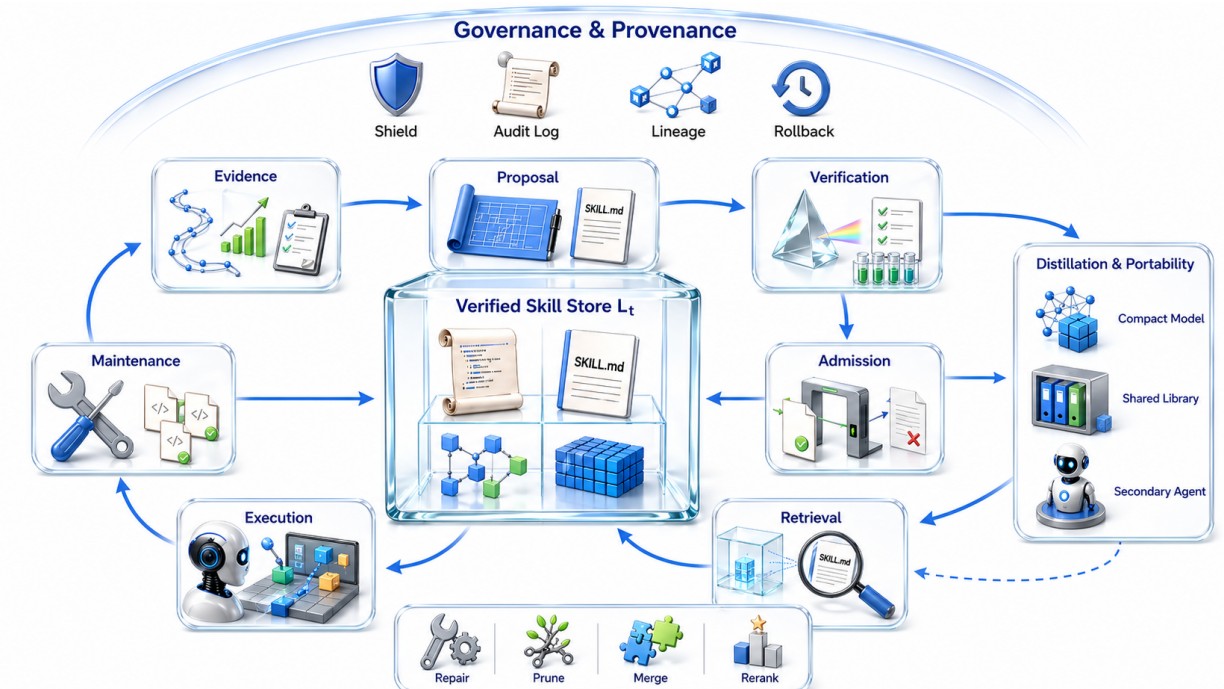

Figure 1: **Dynamic skill systems as lifecycle-managed artifact stores.** Interaction evidence drives proposal and verification; admitted artifacts enter an evolving skill store, where retrieval and execution create further evidence. Maintenance repairs, prunes, merges, or reranks the store over time. Governance and provenance wrap the lifecycle through shields, audit logs, lineage records, and rollback handles, while distillation and portability form a slower side loop. The illustration compresses the textual eight-stage lifecycle: organization/storage is represented by the central store, execution is the action point that produces new evidence, and governance/distillation are wrapper or side-loop functions rather than extra linear stages.

nance can demote, revise, or remove the artifact. Without this lineage, dynamic skills become append-only memories with a more polished file format.

## 5.3 What Counts as Dynamic

We use "dynamic" for systems with a non-trivial library transition $\mathcal{L}_t \rightarrow \mathcal{L}_{t+1}$, not for any system that retrieves a skill at inference time. Retrieval changes the context; dynamic update changes the store. A method can therefore be skill-using without being dynamic, and it can be dynamic in a narrow way if it implements only one transition such as ADD after each task. Lifecycle maturity increases as systems add admission, maintenance, lineage, governance, and two-timescale consolidation.

This distinction sets up the taxonomy in Section 6. The lifecycle table above defines the design commitments a complete dynamic skill system must make; the taxonomy asks which subsets of those commitments each paper actually implements.

| Lifecycle stage | Design question | Main operators | Representative evidence | Characteristic failure |
|---|---|---|---|---|
| Evidence acquisition | What observation justifies a library change? | – | Trajectory/rubric evidence (TRACE2SKILL, SKILLFLOW-BENCH, RAW-EXPERIENCE); web, mobile, embodied, and visual rollouts (ASI, WEBXSKILL, SKILLDROID, EMBODISKILL, XSKILL); reward, failure, or resource signals (SAGE, SKILLRL, MACRO, SKILLFORGE, SKILLFOUNDRY). | noisy evidence; non-stationary utility |
| Proposal / artifact creation | How is evidence converted into an artifact? | ADD, ABSTRACT | Retrospective lessons (ERL, RETROAGENT); executable code and templates (SKILLWEAVER, EVOSKILL, ASI, CONTRACTSKILL, SKILLDROID, HASP); file patches, repository mining, corpus compilation, and contrastive synthesis (SKILLFLOW-BENCH, SKILLREPOMINING, CORPUS2SKILL, SKILLGEN). | task-specific or overgeneral skills |
| Verification and admission | What gate decides whether the candidate enters? | REFINE, REWRITE | Execution checks (SKILLWEAVER, ASI, CONTRACTSKILL, SKILLDROID); Pareto, surrogate, multi-test, rollback, learned, and validation-utility gates (EVOSKILL, SKILLMOO, COEVOSKILLS, SKILLFOUNDRY, PSN, CODE-SHARP, SKILLOPT, SKILLMASTER). | plausible but wrong skills; verifier hacking |
| Organization and storage | Where does the admitted artifact live? | SPLIT, MERGE, RERANK | Capability trees, prerequisite DAGs, invocation/URL graphs, typed ecosystem graphs, structured views, ontologies, dependency retrieval, and scoped packages (AGENTSKILLOS, CODE-SHARP, PSN, WEBXSKILL, SKILLOPS, SSL, SKILLNET, GRAPH OF SKILLS, SKILLDEX). | flat-retrieval collapse; stale structure |
| Retrieval and composition | Which skills affect the next action? | RERANK, COMPOSE | Large-library routing and retrieval (SKILLROUTER, SRA, SKILLFLOW-2025, WILD-SKILLS); graph, typed-DAG, visual, and orchestrated composition (GRAPH OF SKILLS, GRASP, XSKILL, SKILLORCHESTRA); adaptive skill-context construction (SKILLSINJECTOR). | usage without utility; distractor load |
| Maintenance and repair | How is the library kept compact and correct? | PRUNE, MERGE, REFINE, REWRITE | Pruning/merging ablations and local repair (AUTOREFINE, CONTRACTSKILL); recompilation, consolidation, audit, technical-debt diagnosis, maturity gating, optimization, and failure diagnosis (SKILLDROID, XSKILL, AGENTSKILLOS, SKILLOPS, PSN, SKILLFORGE, SKILLFLOW-BENCH, SKILLGRAD, SKILLOPT, EMBODISKILL). | unbounded growth; negative transfer |
| Distillation and portability | What moves between external artifacts, weights, and agents? | DISTILL, ABSTRACT | Fast–slow internalization and adapter-level transfer (METACLAW, K2-AGENT, SKILL0, SKILLSCRAFTER); cross-agent reuse and package compilation (COEVOSKILLS, XSKILL, MUSE-AUTOSKILL, SKILLSMITH). | parametric collapse; compatibility failure |
| Governance and provenance | Can edits be audited, attributed, and rolled back? | logged ADD–PRUNE sequence | Audited graphs and release workflows (ASG-SI, AGENTDEVEL); threat taxonomies, scanners, registry studies, request-conditioned audit, release audit, and semantic supply-chain analysis (SECURE-SKILLS, SKILLSIEVE, AGENTSKILLS-WILD, MALICIOUS-SKILLS-WILD, CREDENTIAL LEAKAGE, MALICIOUS-OR-NOT, STARS, MEDSKILLAUDIT, SKILLSVOTE, SEMANTIC-SUPPLY). | skill injection; leakage; attribution loss |

Table 2: **Lifecycle architecture for dynamic skill systems.** The table is not a method ranking; it identifies the recurring stages at which a system must make design commitments. Later tables specialize this architecture into system families, verification architectures, evidence-graded patterns, and open problems.

## 6 A Lifecycle Taxonomy of Dynamic Skill Systems

The taxonomy asks which lifecycle configuration a system instantiates. We use two levels: lifecycle families for the conceptual map, and seven coding fields in the master coding sheet (Table 11, Appendix A) for auditability.

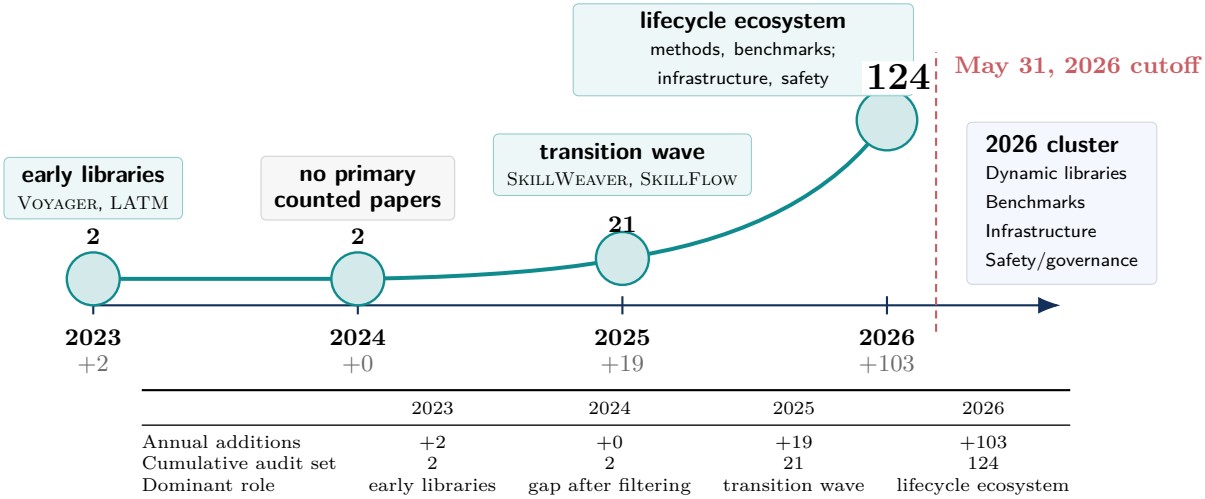

|  | 2023 | 2024 | 2025 | 2026 |
| --- | --- | --- | --- | --- |
| Annual additions | +2 | +0 | +19 | +103 |
| Cumulative audit set | 2 | 2 | 21 | 124 |
| Dominant role | early libraries | gap after filtering | transition wave | lifecycle ecosystem |

Figure 2: **Temporal structure of the dynamic-skills audit set.** Counts summarize the 124 modern papers covered by the survey and exclude only older classical-options and HRL background anchors; boundary/context papers are included for scope but are not treated as primary causal evidence in the evidence-graded patterns. The figure reports annual additions and cumulative coverage through the May 31, 2026 cutoff. Representative work names are milestones rather than an exhaustive bibliography.

### 6.1 Primary Families

Table 3 separates systems that are often conflated. Retrospective lesson systems and PPVH systems both update after a task, but only PPVH has execution-grounded admission. Skill-aware RL and two-timescale systems both use reward, but only the latter separates fast external edits from slow parametric internalization. Cross-user transfer and registry-scale infrastructure both handle many skills, but one concerns distributed authorship and the other routing/storage. Benchmarks such as SKILLFLOW-BENCH and WILD-SKILLS are included because they expose lifecycle behavior method papers often hide.

### 6.2 Seven Coding Fields and Three Couplings

We code each representative system along seven fields: *artifact type* (code, NL lesson, SKILL.md package, memory trace, graph/workflow, or weight delta), *update locus*, *evolution trigger*, *operator repertoire*, *learning signal*, *storage topology*, and *model portability*. These are not claimed to be orthogonal basis dimensions. They are audit fields whose couplings are often the point.

Three couplings matter most. *Artifact–verifier coupling*: executable code supports tests and rollouts, while NL lessons and capability labels rely on weaker downstream or judge-based checks. *Storage–maintenance coupling*: flat stores make ADD cheap but make MERGE and PRUNE costly; graphs and hierarchies expose structure but require more careful rollback. *Trigger–signal coupling*: task-end retrospection naturally supplies critique, RL loops supply reward, user edits supply ownership constraints, and deployment registries supply aggregate utility. This dependence is why Table 3 is the conceptual taxonomy and Table 11 is the coding sheet. Figure 3 gives the corresponding visual summary: families differ less in whether they "use skills" than in which lifecycle stages they actually cover.

| Lifecycle family | Dominant lifecycle stages | Operator footprint | Assurance bottleneck | Representative systems and residual failure |
|---|---|---|---|---|
| Retrospective lesson induction | evidence → proposal → retrieval; task-end updates | ADD, REFINE, ABSTRACT, DISTILL | indirect admission through self-critique or downstream success | ERL, RETROAGENT, EVOLVER, MUSE, XSKILL, SAGER, EMBODISKILL; residual lesson drift, visual-context mismatch, and execution-lapse misdiagnosis |
| Executable PPVH libraries | proposal → verification → admission → execution | ADD, REFINE, PRUNE, COMPOSE | verifier coverage outside sampled contracts | SKILLWEAVER, ASI, WEBXSKILL, CONTRACTSKILL, SKILLDROID, EVOSKILL, COEVOSKILLS, SKILLCRAFT, SKILLFOUNDRY, SKILLFORGE, SKILLMOO, HASP, MACRO; residual verifier narrowness |
| Skill-aware RL systems | evidence/proposal inside the RL loop; optional slow distillation | ADD, REFINE, RERANK, DISTILL | verifier hacking and reward–skill mismatch | SAGE, SKILLRL, AGENTEVOLVER, CODE-SHARP, SKILLMASTER, SKILLFLOW-RECURSIVE, TOOL-R0, COS-PLAY; residual reward-skill mismatch |
| Graph / hierarchy systems | organization → maintenance → retrieval/composition | SPLIT, MERGE, COMPOSE, RERANK, PRUNE | rollback and structural-validation cost | PSN, CODE-SHARP, AGENTSKILLOS, SKILLOPS, SKILLX, SKILLNET, GRAPH OF SKILLS, GRASP, SKILLGRAPH; residual structure decay and technical debt |
| Cross-user and registry-sharing systems | portability → retrieval → governance across users or agents | ADD, REFINE, MERGE, RERANK, PRUNE | ownership, privacy, retrieval quality, and compatibility checks | SKILLCLAW, AUTOSKILL, SKILLFLOW-2025, SKILLSVOTE, SRA, AGENTDEVEL, EVOAGENT; residual dominant-user bias, leakage, retrieval drift, and compatibility failure |
| Skill optimization systems | execution evidence → textual update → validation gate | REFINE, REWRITE, ABSTRACT | local validation may miss transfer regressions | SKILLOPT, SKILLGRAD, SKILLGEN; residual overfitting to validation splits and optimizer-state cost |
| Two-timescale internalization | fast external store → slow parametric or shared consolidation | DISTILL, ABSTRACT, PRUNE, RERANK | quality of the distillation window and probe | METACLAW, K2-AGENT, SKILL0, SKILLSCRAFTER, SELAUR; residual parametric collapse and poor portability |
| Lifecycle benchmarks | measurement of proposal, patching, retrieval, use, and repair | measures ADD, REFINE, PRUNE, usage, and quality gaps | evaluator realism and global-library validity | SKILLFLOW-BENCH, SWE-SKILLS-BENCH, SRA, WILD-SKILLS, SKILLS-BENCH, SKILLLEARNBENCH, RAW-EXPERIENCE, HARMFULSKILLBENCH; reveals lifecycle failures but does not solve them |
| Governance and safety systems | admission/invocation checks plus provenance and release review | PRUNE, quarantine, audit, rollback | attack coverage and defended-system integration | ASG-SI, CLAWSAFETY, AGENTSKILLS-WILD, MALICIOUS-SKILLS-WILD, SEMANTIC-SUPPLY, USER-COMPREHENSION, SKILL-INJECT, SKILLJECT, SKILLSIEVE, STARS, SKILLDEX, MEDSKILLAUDIT, MALICIOUS-OR-NOT; residual sparse integration with live dynamic systems |

Table 3: **Primary lifecycle taxonomy of dynamic skill systems.** Rows are families, not rankings. The table compresses each family into its dominant lifecycle stages, typical operator footprint, assurance bottleneck, and residual failure mode. This makes the comparison sharper than a per-paper list: families differ mainly in which parts of $\mathcal{L}_t \to \mathcal{L}_{t+1}$ they make cheap, verified, or governable.

## 6.3 Master Coding Table

Table 11 in Appendix A instantiates the seven coding fields for representative systems. The "headline" column is factual rather than evaluative so that the table remains a map, not a comment collection. We place the full coding sheet in the appendix because it is an audit artifact; the main text uses Table 3 and Figure 3 for synthesis.

## 6.4 What the Taxonomy Reveals

Three diagonals matter most. Artifact and storage co-vary; trigger and signal co-vary; and lifecycle maturity is visible from the operator set. The newest infrastructure and safety papers add a fourth diagonal: once skills are packaged for registries, scanners, package managers, and repository-context checks become part of the same taxonomy as routing and maintenance. These diagonals motivate the mechanism synthesis of

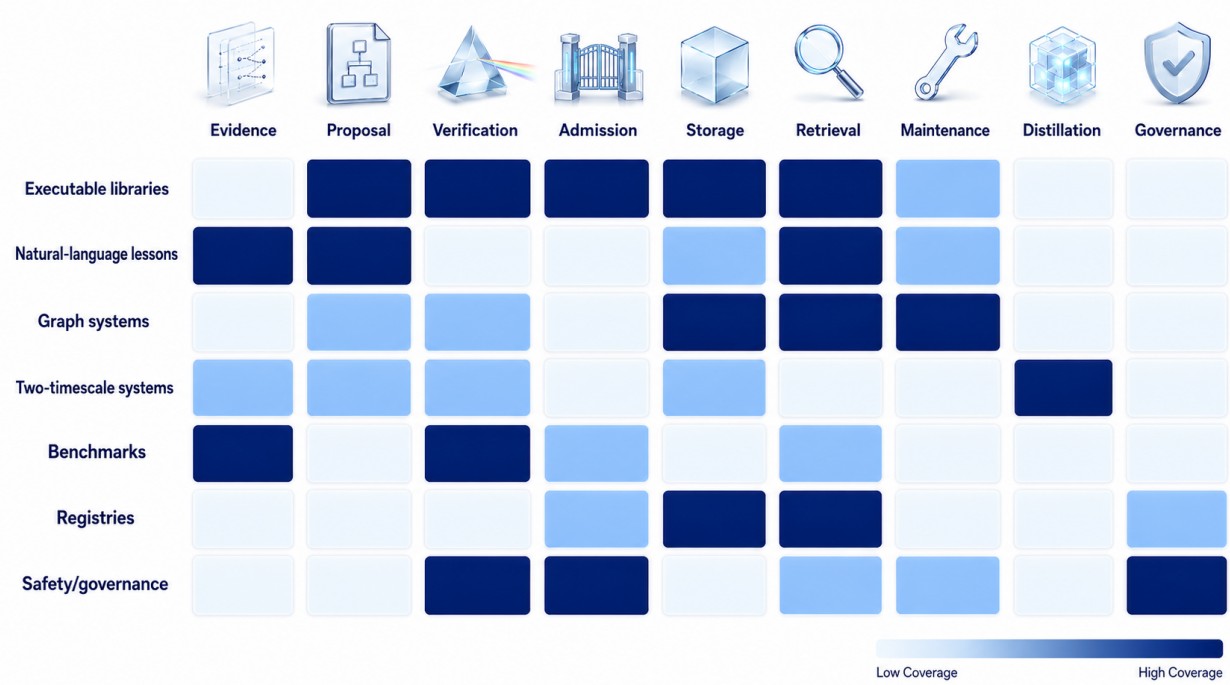

Figure 3: **Lifecycle coverage across dynamic-skill system families.** Cell intensity summarizes how centrally each family implements or evaluates a lifecycle stage in the coding sheet. The figure is a synthesis device rather than a ranking: executable libraries concentrate around proposal, verification, admission, storage, and retrieval; graph systems concentrate around storage, retrieval, and maintenance; two-timescale systems concentrate around distillation; benchmarks expose evidence and verification protocols; registries emphasize storage and retrieval; and safety/governance systems concentrate on admission, verification, and provenance.

§7: operator vocabulary, verification architecture, and fast–slow adaptation. They should be read as coded patterns in a heterogeneous corpus, not as causal conclusions.

# 7 Mechanisms: How Dynamic Skill Stores Improve

Sections 5–6 define the object of study and classify the corpus. This section asks a narrower question: what mechanism turns experience into a better skill store? Across the literature, four choices are load-bearing. A system must decide which edits it can make, how candidate edits are admitted, how the resulting store is organized and retrieved, and when external skills should be consolidated into slower parametric or shared stores. These choices are coupled: a wide edit repertoire requires stronger verification; a large store requires stronger routing and maintenance; and distillation is useful only when the fast-loop library is already selective enough to provide a clean training signal.

## 7.1 Edit repertoire: expansion, compression, and refactoring

The operator vocabulary of Equation 4 is most useful as a diagnostic for library maturity. The simplest systems are *expansion-only*: they add or refine skills after a task, as in Voyager, LATM, SAGE, ERL, and RetroAgent (Wang et al., 2023; Cai et al., 2023; Wang et al., 2025a; Allard et al., 2026; Zhang et al., 2026g). Stronger executable variants already pair Add with abstraction or composition: ASI turns successful web trajectories into verified program actions, while WebXSkill and SkillDroid compile trajectory fragments into reusable web or mobile GUI skill templates (Wang et al., 2025b; 2026g; Chen et al., 2026b). This is enough for short horizons, but it makes the library monotone unless later stages remove, merge, or repair stale artifacts.

Mature systems add a second class of operators that compress or discipline the store. Prune, Merge, and Rerank appear in Wild-Skills, SkillRouter, AutoRefine, Trace2Skill, and AgentSkillOS; SkillOps (Song et al., 2026b) makes the same point explicit as library-time technical-debt management through merge, repair, retire, validator insertion, and adapter insertion. The recurring lesson is that a dynamic library needs a negative operator once distractor load matters. Growth alone is not learning. The mechanism-level question is therefore not only how a system writes skills, but how it removes, merges, or demotes them when later evidence says they are unhelpful.

The late-May 2026 wave adds a sharper specialization: *skill optimization.* SkillOpt (Yang et al., 2026b) treats a skill document as external state optimized by bounded add/delete/replace edits under held-out validation; SkillGrad (Wang et al., 2026b) casts failed and contrastive-successful executions as text gradients plus momentum over a structured skill package; and SkillGen (Ma et al., 2026a) verifies a synthesized skill by its net interventional effect, including both repaired failures and induced regressions. These papers do not introduce a new lifecycle stage, but they make the write-time transition $\mathcal{L}_t \to \mathcal{L}_{t+1}$ look less like reflection and more like optimization over an editable artifact.

A third class changes structure rather than content. PSN applies rollback-gated refactors over an invocation graph; CODE-SHARP mutates a prerequisite DAG; ContractSkill compiles loose web skills into contracts with local patch sites; SkillX splits hierarchies; CoEvoSkills abstracts multi-file `SKILL.md` packages; and Bilevel-MCTS and SkillMOO optimize package structure and pass-rate/cost trade-offs (Shi et al., 2026; Bornemann et al., 2026; Lu et al., 2026b; Wang et al., 2026a; Zhang et al., 2026d; Huang et al., 2026a; Gong et al., 2026). Structural edits are powerful because they can change reuse pathways, not just local skill text, but they also require lineage $\prec$ and rollback because a bad rewrite can invalidate many downstream invocations.

Parametric methods occupy a different regime. Their visible operator set often collapses to {Distill, Abstract}: Refine becomes more training, Merge becomes weight-space interpolation, and Prune is no longer a surgical library edit. This is why SkillsCrafter, MetaClaw, SKILL0, and K2-Agent are best read as two-timescale systems rather than ordinary library editors (Wang et al., 2026f; Xia et al., 2026b; Lu et al., 2026a; Wu et al., 2026b).

## 7.2 Admission and verification are the selection mechanism

Verification is not an auxiliary module; it is the selection pressure that decides which generated artifacts become library state. Table 4 organizes the main verifier forms. Execution verifiers catch runnable failures

through tests, contracts, simulators, or environment rollouts (SKILLWEAVER, ASI, WEBXSKILL, CON-TRACTSKILL, SKILLDROID, EVOSKILL, LIVE-SWE, SKILLFOUNDRY, SKILLCRAFT). Judge-LLM verifiers assess semantic quality, pairwise preference, failure diagnosis, or release readiness (TRACE2SKILL, CO-EVOSKILLS, AUTOREFINE, AGENTSKILLOS, SKILLFORGE, MEDSKILLAUDIT). Rollback and audited-graph verifiers turn verification into a state-transition check, as in PSN and ASG-SI. Utility-based verifiers defer part of the decision to downstream use, as in WILD-SKILLS, SWE-SKILLS-BENCH, EFFISKILL, SKILLGEN, SKILLOPT, and SKILLMASTER.

The important distinction is what each verifier can observe. Execution gates are precise but narrow: passing one contract does not prove broad transfer. Judge gates are broader but vulnerable to evaluator drift and rubric hacking. Rollback gates directly measure behavioral regression, but only on the probe set. Utility gates are cheap and deployment-realistic, but they may admit harmful or misleading skills before enough evidence accumulates. These limitations explain why high-capacity systems increasingly use staged admission: a candidate can be tentative, quarantined, promoted, demoted, or rolled back rather than simply accepted or rejected.

| Form | Timing[a] | Admission[b] | Cost | Coverage | Representative methods |
|---|---|---|---|---|---|
| Execution tests / rollouts | W | hard | high | narrow[c] | VOYAGER, LATM, SKILLWEAVER, ASI, WEBXSKILL, CONTRACTSKILL, SKILLDROID, EVOSKILL, LIVE-SWE, SKILLCRAFT, SKILLFOUNDRY, MUSE-AUTOSKILL, METASURFACE, MACRO |
| Judge-LLM rubric / meta-agent | W + M | hard / Pareto | medium | medium | TRACE2SKILL, COEVOSKILLS, AUTOREFINE, AGENTSKILLOS, SKILLNET, SKILLORCHESTRA, CODE-SHARP, AGENTFACTORY, SKILLFORGE, SKILLGRAD, MEDSKILLAUDIT[d] |
| Ensemble voting | W + M | maturity-gated | medium | wide | AGENTIC-PROPOSING[e] |
| Symbolic / audited | W + M | graph integrity | medium | wide | ASG-SI, SKILLOPS |
| Rollback validation | W + M | Δ-success ≤ 20% | medium | behavioral | PSN |
| Economic / utility | I + eviction | utility threshold | low | deferred | WILD-SKILLS, EFFISKILL, SKILLGEN, SKILLOPT, SKILLMASTER, SKILLSVOTE |
| Security triage / red-team | W + I | quarantine / review | medium | adversarial | AGENTSKILLS-WILD, MALICIOUS-SKILLS-WILD, SKILL-INJECT, SKILLJECT, SKILLSIEVE, STARS, CLAWSAFETY, SKILLATTACK, SUPPLY-CHAIN-POISONING, SEMANTIC-SUPPLY, BADSKILL, CREDENTIAL LEAKAGE |

[a] Timing codes: **W** = write-time, **I** = invocation-time, **M** = maintenance-time (periodic sweep over the library).
[b] Admission policies: *hard* thresholding = admit iff verifier returns success; *Pareto* = admit iff candidate dominates existing skills on a vector of criteria; *maturity-gated* = candidate enters as trial, promoted after usage budget; *graph integrity* = admit iff graph-level audit predicates hold (ASG-SI); Δ-*success* ≤ 20% = tentative admission reverted if task-success rate on recent tasks drops by more than 20% (PSN); *stability-gated* = admit iff fast-loop performance variance is below threshold; *utility threshold* = defer admission, evict below utility floor.
[c] Execution verification is "narrow" because a passing unit test or rollout does not imply generalisation; this is partially why execution-verified methods cap their operator repertoire at {ADD, REFINE, PRUNE} (§7.2).
[d] AGENTFACTORY's verifier is a meta-agent inspection of proposed subagents against a design specification; we classify it here as a judge-LLM variant (rather than as execution) because it inspects the proposal rather than running it against unit-test-style contracts.
[e] AGENTIC-PROPOSING uses a three-LLM-judge ensemble with majority voting. AGENT0 and ASG-SI are driven by curriculum-reward and audit-gated reward respectively; they are not verifier ensembles in the sense used in this row and therefore do not appear here.

Table 4: **Verification architectures in the surveyed dynamic-skill methods.** Three axes organize the space: verifier *form*, verification *timing*, and *admission policy*. The two rightmost columns record the qualitative cost per candidate skill and the coverage over the class of defects each form can catch.

## 7.3 Storage and retrieval control the scaling regime

After admission, the bottleneck shifts from writing good skills to finding the right skills without importing distractors. Flat retrieval is attractive because it is simple, but the surveyed scaling studies repeatedly show a moderate-library-size drop. SINGLE-AGENT-SKILLS reports a sharp decline beyond roughly 64–128 skills when many agent skills are compiled into a single-agent library; WILD-SKILLS shows that realistic distractors can erase apparent gains from curated skills; SKILLROUTER shows that 80K-scale registries require full-text retrieve-and-rerank rather than metadata-only selection; SRA shows that retrieval, incorporation, and

downstream utility must be evaluated separately because agents may load skills at similar rates regardless of whether the gold skill is present or needed; and SKILLSINJECTOR (Li et al., 2026d) shows that even a fixed library needs adaptive selection, budgeting, and set-aware description rendering because static all-injection can degrade performance (Li, 2026; Liu et al., 2026h; Zheng et al., 2026; Su et al., 2026).

The response is to make storage reflect reusable structure. Hierarchies and capability trees support audit and specialization (AGENTSKILLOS, SKILLX, CORPUS2SKILL, UNI-SKILL); DAGs encode prerequisites or typed composition (CODE-SHARP, GRASP); invocation and relation graphs support refactoring, dependency-aware retrieval, and provenance (PSN, GRAPH OF SKILLS, SKILLGRAPH, WEBXSKILL, ASG-SI) (Nie et al., 2026); typed ecosystem graphs expose dependency, compatibility, redundancy, and alternative edges for library health diagnosis (SKILLOPS); structured intermediate representations make retrieval and audit less dependent on raw prose (SSL) (Liang et al., 2026a); compilation and boundary extraction make runtime interfaces smaller (SKILLSMITH) (Xu et al., 2026); ontologies and registries support portability and package-level governance (SKILLNET, SKILLDEX, SKILLSVOTE) (Liu et al., 2026c). The mechanism-level insight is that retrieval quality is partly an indexing problem and partly a maintenance problem. Once a store grows, storage topology, reranking, pruning, and provenance become one system.

### 7.4 Update clocks separate adaptation from consolidation

Many recent systems separate a fast external loop from a slow internal loop. The fast loop edits files, procedures, code snippets, or graph nodes after tasks; the slow loop distills selected behavior into adapters, weights, shared libraries, or compact package formats. This separation lets the agent learn quickly without paying the cost or risk of continuous model updates, while still allowing stable skills to become cheaper to invoke later.

Table 5 compares the recurring decouplings. Some systems share the artifact between loops (METACLAW, K2-AGENT); others share reward or critique signals (SAGE, AGENTEVOLVER, TOOL-R0, COS-PLAY); others share curricula or verifiers (AGENT0, SCALAR, ASG-SI). The key design variable is promotion: prevalence is cheap, reward is task-grounded, coverage favors behavioral diversity, and stability is closest to measuring whether distillation will preserve rather than corrupt a skill.

Two-timescale adaptation is not automatically superior. A noisy fast-loop library gives the slow loop noisy targets, so distillation can compress mistakes as well as discoveries. Conversely, an over-conservative fast loop leaves useful knowledge external and expensive. The open variable is the consolidation schedule: when to distill, what to distill, and whether the slow-loop result should replace, augment, or merely rerank the external library.

| Method | Mode[a] | Fast artifact | Slow artifact | Promoted on | Slow trigger | Slow vs. fast |
|---|---|---|---|---|---|---|
| K2-AGENT (Wu et al., 2026b) | SA | SKILL.md + case | LoRA / adapter | reward ∩ prevalence | task end | augments |
| METACLAW (Xia et al., 2026b) | SA | SKILL.md | LoRA | prevalence | periodic | augments |
| AGENTEVOLVER (Zhai et al., 2025) | SS | NL heuristic | policy weights | RL replay buffer | RL update | augments |
| TOOL-R0 (Acikgoz et al., 2026) | SS | tool proposals | policy weights | reward (dual self-play) | RL step | augments |
| SAGE (Wang et al., 2025a) | SS | Python skill fn. | policy weights | skill-integrated reward | task end | augments[b] |
| COS-PLAY (Wu et al., 2026a) | SS | skill-bank entries | GRPO adapters | rollout reward + skill utility | co-evolution step | augments |
| AGENT0 (Xia et al., 2025b) | SC | code proposals | executor weights | coverage / curriculum | co-evolution step | augments |
| SCALAR (Zabounidis et al., 2026) | SC | env / skill curation | policy weights | teacher signal | RL step | augments |
| ASG-SI (Huang & Huang, 2025) | SV | audited skill graph | policy weights | audit-gated reward | RL step | augments |
| MEMENTO (Zhou et al., 2026a) | —[c] | SKILL.md + case | (no slow loop) | — | — | fast-only |
| LSE (Chen et al., 2026e) | —[d] | prompt context | prompt + weight | reward | — | fast-only |
| CO-EVOLVING (Jung et al., 2025) | —[e] | code + critique | policy weights | hard-negative pool | batch complete | single-loop |
| AGENTIC-PROPOSING (Jiao et al., 2026) | —[f] | skill proposals | policy weights | ensemble-verifier pass | RL step | single-loop |

[a] Decoupling mode (§7.4): **SA** = shared-artifact (fast and slow loops edit and distill the same textual skill); **SS** = shared-signal (both loops consume the same reward / critique stream at different cadences); **SC** = shared-curriculum (fast loop generates tasks for the slow loop); **SV** = shared-verifier (same verifier gates both loops at different stringency).
[b] SAGE's fast artifact is an executable Python skill function updated under a skill-integrated GRPO reward; the slow loop augments this artifact rather than replacing it, and we flag the row because it is the only **SS** method whose fast-loop artifact is code rather than a natural-language heuristic.
[c] MEMENTO is listed as a fast-only baseline for contrast; it has no slow loop and therefore no decoupling mode applies.
[d] LSE trains a 4B edit policy that emits context-level edits as a learned action; there is no separate fast/slow decomposition because the learned editor *is* the fast loop and its weights are the only parametric artifact, so the four modes do not apply.
[e] CO-EVOLVING runs alternating DPO over a hard-negative trajectory pool; critiques and policy updates live in a single optimization loop, with no separate slow-loop distillation over a persistent skill library.
[f] AGENTIC-PROPOSING pairs a fast loop that proposes *problems* (not library edits) with an RL loop that updates the policy against verifier-gated rewards; because the fast-loop output is not a skill artifact, the two-timescale decoupling modes above do not apply.

Table 5: **Two-timescale decoupling modes in methods with both a fast in-context library and a slow parametric loop.** Rows are keyed by method and grouped by the four decoupling modes the literature converges on. The "Promoted on", "Slow trigger", and "Slow vs. fast" columns characterize what moves between loops, when, and whether the slow-loop output replaces or augments the fast-loop library.

## 7.5 Mechanism-level synthesis

The mechanism picture is compact. Dynamic skill systems improve when expansion is paired with compression, admission is paired with re-verification, retrieval is paired with structure, and fast editing is paired with slower consolidation. The same four requirements explain why apparently different systems occupy coherent regimes: executable-skill systems emphasize execution gates and small edit repertoires; natural-language lesson systems emphasize cheap proposal and later maintenance; graph and hierarchy systems trade storage complexity for retrieval and refactoring; and parametric systems trade editability for amortized inference. The strongest systems are not those with the largest libraries, but those that make the library easier to verify, route, repair, and consolidate over time.

# 8 Evaluation

A benchmark that reports only endpoint task success hides the phenomena that define dynamic skills: skill inflation, incorrect-skill drift, maintenance-off collapse, retrieval degradation as flat libraries grow, and usage without utility. This section audits evaluation through a lifecycle lens: how libraries are created, repaired, routed, compacted, and governed over time.

## 8.1 The Benchmark Landscape

The surveyed papers report on four partly-overlapping benchmark clusters. The first is the *agent-task* cluster: WebArena, VisualWebArena, Mind2Web, AgentBench, ST-Bench, SWE-bench-style command-line environments, and related terminal or software-engineering suites. These benchmarks evaluate an agent end-to-end on held-out tasks; dynamic-skill papers often inherit the benchmark and report a single success rate. The second is the *code-execution* cluster: HumanEval, MBPP, LiveCodeBench, APPS, and software-engineering task suites with intrinsic execution verification. These benchmarks are why code-skill and skill-aware RL papers can afford stronger admission gates. The third is the *reasoning* cluster: AIME, MATH, GPQA, and HLE; this is where AGENTIC-PROPOSING and MEMENTO report some headline results, and where verifier quality is often load-bearing because ground truth is unambiguous.

The fourth cluster is most specific to this survey: *skill-lifecycle evaluation.* SKILLSBENCH (Li et al., 2026c) measures skill usefulness across tasks, SWE-SKILLS-BENCH (Han et al., 2026) isolates the marginal value of public SWE skills under deterministic tests, WILD-SKILLS (Liu et al., 2026h) stresses retrieval realism under distractors and 34K-candidate retrieval, SINGLE-AGENT-SKILLS (Li, 2026) gives the clearest controlled size sweep for 64–128-skill flat-retrieval degradation, SKILLROUTER (Zheng et al., 2026) evaluates full-text routing over an 80K skill pool, and SRA (Su et al., 2026) decomposes large-corpus skill augmentation into retrieval, incorporation, and end-task utility over a 26,262-skill corpus. SKILLFLOW-BENCH (Zhang et al., 2026j) is the most lifecycle-aligned benchmark: 166 runnable tasks across 20 DAEF-structured families, empty initial family libraries, JSON skill patches from trajectories, and reports of completion, turns, cost, output tokens, final skill count, and skill-use rate. Its Table 1 gives the key finding that skill use is not skill utility: Claude Opus 4.6 improves from 62.65% to 71.08%, while Kimi K2.5 gains only +0.60 points despite 66.87% skill use and GPT 5.3 Codex regresses by 6.02 points.

SKILLLEARNBENCH (Zhong et al., 2026) asks whether agents can continually generate useful skills, not merely use supplied ones, and finds a large gap to human performance across 20 verified skill-dependent tasks. RAW-EXPERIENCE (Huang et al., 2026c) broadens this into a lifecycle study over experience generation, skill extraction, and skill consumption, finding that model-generated skills help on average but can transfer negatively and that a strong extractor need not be a strong consumer. SKILLGEN (Ma et al., 2026a) gives the interventional version of the same evaluation idea: candidate skills are selected by net effect, counting both repaired failures and new regressions. SWE-SKILLS-BENCH gives the complementary negative result for supplied skills: across 49 public SWE skills and about 565 tasks, average pass-rate gain is only +1.2%, 39 skills yield no improvement, and three regress. HARMFULSKILLBENCH (Jiang et al., 2026c) is the safety analogue, separating harmful skill presence from explicit and implicit invocation. MEDSKILLAUDIT (Hou et al., 2026) adds a domain-release protocol: evaluate the reusable skill artifact itself for release readiness.

| Benchmark | Primary lifecycle coverage | Self-gen. | Revision | Trajectory | Usage utility | Main limitation for this survey |
|---|---|---|---|---|---|---|
| SKILLSBENCH | skill usefulness under provided or generated skills | ✓ | – | – | – | evaluates skill effectiveness more than longitudinal library management |
| SWE-SKILLS-BENCH | marginal utility of public SWE skills under fixed repositories and deterministic tests | – | – | – | ✓ | strong paired utility evidence, but not a dynamic self-repair protocol |
| WILD-SKILLS | realistic retrieval, distractors, curation utility over a 34K pool | – | – | ~ | ✓ | strong retrieval realism, but not sequential self-repair |
| SKILLROUTER | full-text routing at scale over an 80K skill pool | – | – | – | ~ | measures retrieval accuracy/speed more than downstream lifecycle utility |
| SRA-BENCH | retrieval, incorporation, and end-task utility over a 26K-skill corpus | – | – | – | ✓ | decomposes scalable retrieval but does not evaluate generation or maintenance |
| RAW-EXPERIENCE | experience generation, skill extraction, and skill consumption across domains | ✓ | – | ✓ | ✓ | systematic lifecycle study, but not a deployed evolving store |
| SKILLGEN | net-effect verification of synthesized skills as interventions | ✓ | ✓ | ~ | ✓ | single-skill synthesis rather than large-library maintenance |
| GRAPH OF SKILLS | dependency-aware retrieval over 200–2,000 skill libraries | – | – | – | ✓ | strong structural retrieval evidence, but no skill repair loop |
| LIFELONGAGENTBENCH | lifelong task sequences and accumulation pressure | ~ | ~ | ✓ | – | adjacent lifelong-learning benchmark, not skill-artifact-specific |
| PROEVOLVE | programmable distribution shift for evolving agent benchmarks | – | – | ✓ | – | supplies drift protocol but not a skill-library protocol |
| SKILLFLOW-BENCH | discovery, JSON skill patches, repair, reuse, and compactness across 166 tasks | ✓ | ✓ | ✓ | ✓ | family reset avoids global heterogeneous-library retrieval; model and harness are co-varied |
| SKILLSVOTE | collection, recommendation, skill-linked credit, and evidence-gated evolution | ✓ | ✓ | ✓ | ✓ | reports governed evolution, but full open-ecosystem replication is hard |
| SKILLLEARNBENCH | continual skill generation on 20 verified skill-dependent tasks | ✓ | ✓ | ✓ | ✓ | evaluates generation quality, but task count is still small |
| CLAWSAFETY | safety surfaces for skill injection in personal agents | – | – | – | ✓ | attack benchmark, not a defended dynamic-skill evaluation |
| HARMFULSKILLBENCH | harmful skill prevalence and explicit/implicit unsafe invocation | – | – | – | ✓ | safety benchmark; not a repair or mitigation protocol |
| MEDSKILLAUDIT | domain-specific release-readiness audit for medical research skills | – | ✓ | – | ~ | reliability study over 75 skills, not downstream task benchmark |

Table 6: **Benchmark coverage over dynamic-skill lifecycle dimensions.** The table separates benchmark roles rather than ranking benchmarks. "Self-gen." means the benchmark evaluates skills produced by the agent; "Revision" means skills can be patched or repaired over time; "Trajectory" means the protocol exposes a time-ordered library or task sequence; "Usage utility" means the benchmark can distinguish reading/calling a skill from actually improving task outcome.

## 8.2 Reported Metric Families

Four metric families dominate. *Terminal success rate* or pass@$k$ remains the default: evaluate after training or after library construction and report a single number. This metric is useful but incomplete because it hides library-size effects, focus effects, and maintenance effects. *Sample efficiency* or wall-clock-to-threshold metrics (EvoSkill, SAGE, K2-Agent, SkillDroid, SkillOpt, SkillGrad) report how quickly a method reaches a target success rate or reduces repeated-use cost; these are better for surfacing the weaker-backbone and repeated-use advantages that dynamic skills often show. *Library-size and retrieval stress tests* (Single-Agent-Skills, Wild-Skills, SkillRouter, Graph of Skills, SRA, SkillsInjector) expose scaling behavior, but only a minority of papers report them. *Lifecycle metrics*, newly visible in SkillFlow-Bench, SkillLearnBench, SWE-Skills-Bench, Raw-Experience, SkillsVote, and SRA, report final skill count, file-kind composition, skill-use rate, generated-skill quality, retrieval-versus-incorporation gaps, and the gap between skill usage and task improvement.

XSkill (Jiang et al., 2026a) adds a multimodal metric design: Average@4 and Pass@4 over repeated rollouts, plus ablations over the skill stream, experience stream, and knowledge managers. It is not a full library-trajectory protocol, but it separates average rollout quality from best-of-four exploration and shows that transfer can raise Pass@4 while lowering Average@4 on weaker open-source backbones.

Two quantities remain under-reported. The first is *operator velocity*: counts of Add, Refine, Merge, Prune, and Rerank per task or per dollar. Without this quantity, the velocity–soundness tradeoff between PSN, CODE-SHARP, SkillMOO, and Bilevel-MCTS cannot be compared quantitatively. The second is *repair quality*: whether a later skill patch actually corrects a faulty abstraction. SkillFlow-Bench exposes this qualitatively through incorrect-skill drift and skill inflation, and SkillForge does so in a deployment-style failure-diagnosis loop, but the field still lacks a standard scalar repair metric.

## 8.3 Four Comparisons That Remain Hard

Benchmark progress does not yet make the literature head-to-head comparable. *Cross-operator comparison* lacks operator velocities. *Cross-library-size comparison* lacks smooth size sweeps. *Cross-backbone comparison* is confounded by model, harness, context length, and tool surface, even in useful early transfer studies such as CoEvoSkills and XSkill. *Cross-task-distribution comparison* remains immature: adjacent protocols such as ELL/StuLife (Cai et al., 2025), LifelongAgentBench (Zheng et al., 2025b), and ProEvolve (Li et al., 2026a) expose lifelong, proactive, or programmable shift, but most deployment papers still use bespoke task streams. These limits determine how the next section treats evidence: the patterns in §9 rely mainly on within-paper ablations and convergent benchmark behavior rather than cross-paper leaderboards.

## 8.4 Toward Trajectory-Aware Evaluation

A trajectory-aware protocol should report the library as a time series. Four elements are sufficient: performance, skill count, and retrieval quality at a grid of task indices or library sizes; operator velocities for Add, Refine, Merge, and Prune; a drift schedule or family-transfer condition; and a maintenance-off or repair-off ablation. Single-Agent-Skills' size sweep, SkillRouter's 80K routing benchmark, SRA's decomposition of retrieval versus incorporation, SWE-Skills-Bench's with-skill/without-skill deltas, SkillFlow-Bench's skill-count trajectories, SkillsVote's skill-linked credit assignment, and Raw-Experience's extractor-consumer split are complementary starting points.

SkillFlow-Bench moves the field in this direction, but its family-reset design leaves open how one global library behaves under heterogeneous workflows. The next step is to add cross-family global-library protocols, operator-velocity logging, and maintenance-off ablations.

# 9 Seven Evidence-Graded Patterns

Building on the evaluation audit in §8, the primary dynamic-skill cluster within the 124-paper modern audit set surfaces seven recurring patterns across methods, benchmarks, and artifact types. They should not be

read as pooled effects. Evidence is strongest for within-paper ablations, moderate for convergent benchmark behavior, and weakest for architectural corroboration without ablation.

Table 7 summarizes the evidence for the section and assigns each pattern the evidence grade defined in §2.5; § 12 uses these patterns to frame the open-problem agenda.

If the analysis is restricted to the primary dynamic-skill systems and excludes memory-only, registry-only, and purely parametric boundary cases, the strongest patterns are admission, verifier quality, and maintenance/repair. Retrieval scaling remains moderately supported because it has both a controlled size sweep and registry-scale corroboration, but it still lacks a shared benchmark across storage designs. Weaker-backbone gains, focused-library advantage, and write-time abstraction are useful cross-system signals rather than settled effects; they are retained because they describe recurring design pressure, not because they support a pooled effect size.

### 9.1 Curated skills outperform unverified self-generated skills

One of the clearest within-paper findings in the surveyed corpus is that *admission matters*. Libraries whose write-time verifier is stronger than "the agent proposed it" usually beat libraries that admit any proposal, and the evidence spans methods at both ends of the verification spectrum. EvoSkill (Alzubi et al., 2026) reports that its Pareto-front selection over held-out validation performance avoids the redundant/conflicting-skill accumulation seen under greedy acceptance and yields +7.3 absolute points on OfficeQA (Table 1 / Figure 2) and +12.1 points on SealQA; CoEvoSkills (Zhang et al., 2026d) provides the cleanest verifier ablation in the corpus, with Table B1 showing that removal of the surrogate verifier drops SkillsBench pass rate from 71.1% to 41.1%; SkillWeaver (Zheng et al., 2025a) reports that removing the "practice + verify" phases of PPVH (leaving only propose+hone) drops benchmark performance below the no-skill-library baseline on its hardest web tasks; and Trace2Skill (Ni et al., 2026) reports that prevalence-weighted consolidation is useful only when consolidation also filters on a judge score. New executable web and GUI systems strengthen the same point: ASI (Wang et al., 2025b) gains over text skills partly because induced programs are execution-verified before becoming actions, WebXSkill (Wang et al., 2026g) reports that validation and curation are necessary for its grounded/guided web-skill gains, and ContractSkill (Lu et al., 2026b) improves self-generated web skills by turning failure into verifier-localized patch admission. Late-May optimization papers sharpen the same mechanism: SkillGen selects candidate skills by net interventional effect, SkillOpt accepts text edits only when held-out validation improves, and SkillMaster trains skill edits with counterfactual utility rewards (Ma et al., 2026a; Yang et al., 2026b;a). Additional 2026 systems strengthen the same point from new domains: SkillFoundry (Shen et al., 2026) only admits scientific packages after contract/provenance/test validation, SkillsVote (Liu et al., 2026c) admits only successful reusable discoveries after skill-linked credit assignment, and SkillForge (Liu et al., 2026e) improves deployed support skills through failure analysis, diagnosis, and optimization rather than blind rewriting. The pattern also appears in maintenance-heavy systems: AutoRefine (Qiu et al., 2026)'s pruning/merging gate and AgentSkillOS (Li et al., 2026b)'s audit-gated organization both report that disabling the gate erodes downstream task success.

**Caveats.** This pattern says that replacing no verification with some verification is usually valuable, not that more verification is always better. The main exception is the memory/trajectory family (SimpleMem, parts of MUSE) (Yang et al., 2025a), where artifacts are episodic cases and admission is closer to relevance filtering than quality control.

### 9.2 Verification *quality* is often decisive in skill-aware RL

Inside skill-aware RL, the *quality of the verifier or reward-shaping signal* is often one of the most load-bearing choices. CoEvoSkills' Table B1 surrogate-verifier ablation isolates a 30-point verifier effect; CODE-SHARP improves success from 24.30% to 41.02% through refinement mutations under a learned gate; Agentic-Proposing's verifier ensemble plus dynamic pruning improves problem validity from 68.7% to 82.3%; Co-Evolving (Jung et al., 2025) attributes substantial gain to hard-negative construction; and SELAUR (Zhang et al., 2026b) shows that uncertainty-aware reward shaping can matter as much as the RL algorithm. SkillMaster adds the clearest new variant: skill edits receive counterfactual probe utility and

| Observed pattern | Grade | Primary supporting papers | Key caveat / evidence boundary |
|---|---|---|---|
| Admission gates matter: curated skills beat unverified self-generated skills. | A/B | EvoSkill, CoEvoSkills, SkillWeaver, ASI, WebXSkill, ContractSkill, Trace2Skill, SkillGen, SkillOpt, SkillMaster, SkillsVote, SkillFoundry, SkillForge, AutoRefine, AgentSkillOS | Case-based memory (SimpleMem, parts of MUSE) where admission = retrieval relevance; hard admission filters can remove rare-task cases. |
| In skill-aware RL, verifier *quality* is often one of the most decisive engineering choices. | B | CoEvoSkills, CODE-SHARP, Agentic-Proposing, Co-Evolving, SELAUR, SkillMaster, SkillFlow-Recursive | ASG-SI is architectural corroboration, not benchmark-strength ablation evidence; executable-library regimes with strong execution verifiers can see diminishing returns. |
| Flat retrieval often drops at moderate library scale (roughly one hundred skills). | B/C | Single-Agent-Skills, Wild-Skills, AgentSkillOS, SkillRouter, SRA, Graph of Skills, SkillsInjector | Single-Agent-Skills supplies the controlled 64–128-skill curve; SkillRouter and SRA supply large-corpus routing/incorporation evidence rather than smooth size sweeps. |
| Several studies report larger relative gains for weaker backbones. | C | SkillWeaver, MetaClaw, EvoSkill, Agentic-Proposing | Libraries of rare-task specializations can invert the effect: strong models route reliably to them while weaker models cannot; XSkill shows transferred knowledge can improve Pass@4 while hurting Average@4 on weaker backbones. |
| Focused libraries often beat comprehensive ones even when the focused one is a subset. | B/C | SkillX, Wild-Skills, CASCADE, SkillMOO, SWE-Skills-Bench, SkillLearnBench, SkillFlow-Bench, SkillsInjector, Raw-Experience, SKILL.md-trimming literature | Non-stationary deployment (SkillClaw, AutoSkill, MetaClaw): a focused library staleness-dominates; the result becomes an argument for fast Prune. |
| At moderate-to-large library sizes, maintenance becomes load-bearing. | B | AutoRefine (TravelPlanner maintenance ablation: $35.6 \rightarrow 31.1$, $4.5\times$ repository growth, $0.71 \rightarrow 0.08$ utilization), SkillOps, ContractSkill, SkillDroid, EmbodiSkill, AgentSkillOS, Wild-Skills, EffiSkill, PSN, SkillFlow-Bench, SkillGrad, SkillOpt, MetaClaw, SKILL0 | Very short task horizons (a handful of tasks pre-evaluation) where maintenance cost cannot amortize; SkillFlow-Bench is benchmark corroboration, not a maintenance-off ablation. |
| Write-time abstraction (Abstract/Distill at authoring) is usually the stronger backbone than read-time abstraction alone. | B/C | Trace2Skill, CASCADE, SimpleMem, XSkill, SkillFlow-Bench, SkillGen, SkillOpt, SkillGrad, Raw-Experience | SimpleMem and XSkill are adjacent memory / dual-store evidence; SkillFlow-Bench is a structured artifact-creation control rather than a clean write/read ablation; SkillTTA shows read-time transient skills can still help. |

Table 7: **Seven evidence-graded patterns in the dynamic-skills literature (2023–2026).** Each row summarizes an observed pattern (§9), an evidence grade using the convention in §2.5, the methods that provide supporting ablations or measurements, and the main evidence boundary. Grades are deliberately conservative: they reflect heterogeneous backbones, task surfaces, evaluators, and artifact types rather than pooled effect sizes. The strongest themes concern write-time discipline; the retrieval-scaling and focus rows argue for smaller libraries from the retrieval-resolution and distractor-load ends respectively; the weaker-backbone row is suggestive and deployment-facing. The open-problem agenda in §12 is organized around these patterns.

separate advantage normalization from task actions (Yang et al., 2026a). SKILLFLOW-RECURSIVE (Zhang et al., 2026f) is architectural corroboration from flow matching, where recursive skill evolution uses trajectory-balance diagnostics rather than direct prompt judgment. ASG-SI (Huang & Huang, 2025) specifies the audited graph and verifiable-reward machinery a deployed system would need.

**Caveats.** This pattern is specific to skill-aware RL. In executable libraries, once execution verification exists, additional rollouts or tighter contracts may have diminishing returns relative to PRUNE and RERANK.

### 9.3 Flat retrieval often degrades at moderate library sizes

Flat skill libraries can show an accuracy drop at moderate sizes, consistent with a top-$k$ retrieval signal-to-noise collapse. SINGLE-AGENT-SKILLS (Li, 2026) gives the clearest controlled size sweep: selection remains high at 16–64 skills, degrades around 128 skills, and drops sharply at 256 skills unless hierarchical routing or disambiguating instructions are added. AGENTSKILLOS (Li et al., 2026b) motivates its capability tree with flat-invocation collapse, WILD-SKILLS (Liu et al., 2026h) shows the same mechanism under forced loading, autonomous selection, distractors, 34K-pool retrieval, and no curated task-specific skills, and SKILLROUTER (Zheng et al., 2026) shows that 80K-scale registries need full-text retrieve-and-rerank because metadata-only routing loses implementation-level signals. SRA (Su et al., 2026) adds a decomposed 26K-corpus benchmark: better retrieval helps, but incorporation and need-aware loading remain separate bottlenecks. GRAPH OF SKILLS (Liu et al., 2026a) adds 200–2,000-skill evidence that dependency-aware graph retrieval can preserve reward while compressing token use. SKILLSINJECTOR gives the read-time analogue: static all-injection can collapse as candidate pools grow, while adaptive budgeting and set-aware rendering improve downstream pass rates (Li et al., 2026d).

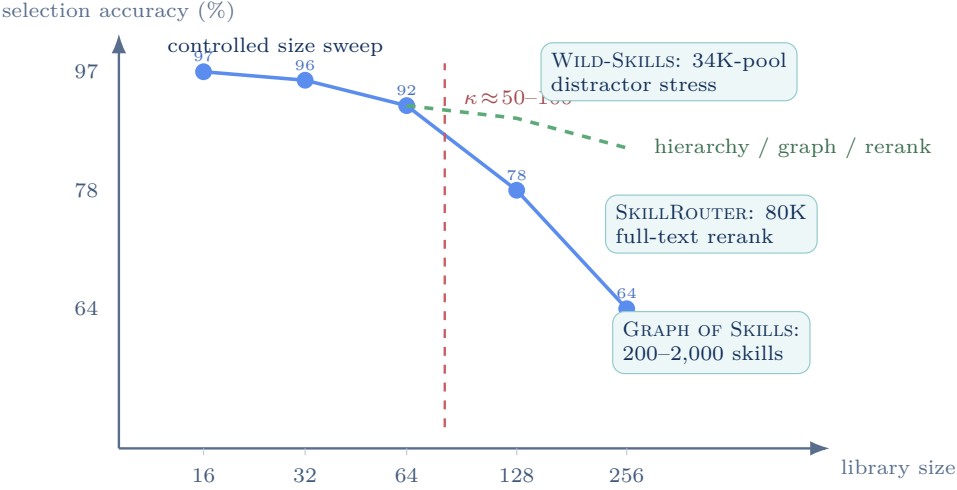

Figure 4: **Retrieval-scaling evidence behind R3.** The blue curve plots the controlled SINGLE-AGENT-SKILLS sweep reported in the paper: 16–32 skills remain around 96–98%, 64 skills at 92%, 128 skills at 78%, and 256 skills at 64%. The dashed green path summarizes the mitigation family rather than a pooled leaderboard: hierarchy, graph retrieval, and full-text reranking push the failure rightward in SINGLE-AGENT-SKILLS, WILD-SKILLS, SKILLROUTER, and GRAPH OF SKILLS, but they are not yet comparable under one shared harness.

**Caveats.** The location of the drop depends on retriever quality, description length, skill orthogonality, and whether the corpus exposes full skill bodies. Hierarchical, DAG, and ontology storage can push the drop rightward (AGENTSKILLOS, SKILLX, SKILLORCHESTRA) (Wang et al., 2026c); full-text reranking (SKILLROUTER) addresses the complementary registry-scale routing problem (Zheng et al., 2026). The literature has not shown general removal.

### 9.4 Several studies report larger relative gains for weaker backbones

Several studies report larger relative gains for weaker base models than for stronger ones, but the evidence is convergent rather than a controlled cross-paper effect. SKILLWEAVER (Zheng et al., 2025a) closes a larger fraction of the capability gap on weaker web-agent backbones; METACLAW (Xia et al., 2026b) reports larger relative two-timescale gains on weaker backbones in its setting; EVOSKILL reports analogous gap compression; and AGENTIC-PROPOSING's headline result is on a 30B model, with a smaller relative lift on the frontier teacher. Because these papers vary in harness, context budget, and gain definition, we treat the pattern as a deployment-facing hypothesis rather than as a comparable effect-size estimate.

**Caveats.** The effect depends on library content. Common-but-not-obvious heuristics help weaker models more; rare specializations can invert the pattern if weaker models cannot route to them reliably. XSKILL's Qwen transfer results are the caution: transfer can raise Pass@4 by encouraging exploration while lowering Average@4.

### 9.5 Focused libraries often beat comprehensive ones

A library curated to a narrow task distribution can outperform a more comprehensive library, even when the former is a subset of the latter. This is the content-level dual of retrieval degradation: the failure mechanism is distractor load. SKILLX (Wang et al., 2026a) reports that task-family-tuned hierarchies beat flat libraries; WILD-SKILLS shows that retrieval-utility filtering beats skills that looked useful in isolation; and CASCADE (Huang et al., 2025) reports that consolidation beats pure growth at fixed compute. SKILL-MOO (Gong et al., 2026) optimizes pass rate and cost jointly, SKILLLEARNBENCH (Zhong et al., 2026) shows that plausible skill text does not imply competence, SWE-SKILLS-BENCH (Han et al., 2026) reports only +1.2% average gain across public SWE skills with many zero-utility or negative cases, and SKILLFLOW-BENCH (Zhang et al., 2026j) finds that stronger settings maintain compact revised skills while weaker settings often fragment into many low-utility files. SKILLSINJECTOR and RAW-EXPERIENCE add the context and extraction views: a larger injected set can hurt, and a skill useful for one consumer can transfer negatively to another (Li et al., 2026d; Huang et al., 2026c). The single-skill analogue is the SKILL.md focused-prompt result: trimming L2 prose to task-relevant instructions improves use.

**Caveats.** In non-stationary deployments (SKILLCLAW, AUTOSKILL, METACLAW), focused libraries can become stale; the result then argues for fast PRUNE, not permanent narrowness.

### 9.6 At moderate-to-large library sizes, maintenance becomes load-bearing

Systems with at least one negative or repair operator—PRUNE, MERGE, REFINE, or an equivalent—often beat monotonic-growth systems once distractor load matters. AUTOREFINE (Qiu et al., 2026) reports the headline ALFWorld result in its Table 1, but its maintenance-off evidence is the TravelPlanner validation ablation: removing periodic pruning and merging lowers final pass rate from 35.6% to 31.1%, grows the repository by 4.5×, and reduces utilization from 0.71 to 0.08 (its Figures 2–3). SKILLOPS (Song et al., 2026b) gives a direct library-time variant: removing library-time maintenance drops standalone ALFWorld success to 71.9%, while the full system reports 79.5% and plug-in gains of +0.68–+2.90 points for retrieval-heavy baselines. CONTRACTSKILL shows the single-artifact analogue: local verifier-guided patch repair beats naive self-generated web skills on VisualWebArena (Lu et al., 2026b). SKILLDROID shows the repeated-use analogue: failure-count recompilation lets a mobile GUI skill library improve while a stateless baseline degrades under instruction variation (Chen et al., 2026b). EMBODISKILL adds an embodied repair caveat: the system must distinguish skill defects from execution lapses before rewriting valid guidance (Ju et al., 2026). AGENTSKILLOS attributes scaling to periodic capability-tree audit; WILD-SKILLS and EFFISKILL (Wang et al., 2026i) find utility-based filtering can beat increased admission; and PSN's maturity gating loses its stated advantage when disabled. SKILLFLOW-BENCH adds that weaker settings often fail by skill inflation and incorrect-skill drift rather than inability to write skills. SKILLGRAD and SKILLOPT frame repair as iterative text-space optimization rather than one-shot reflection (Wang et al., 2026b; Yang et al., 2026b).

Parametric systems express the same point through distillation windows (METACLAW, SKILL0) (Lu et al., 2026a): mistimed consolidation lets the fast-loop library grow without bound.

**Caveats.** The exception is very short horizons, where maintenance cost has no time to amortize.

### 9.7 Write-time abstraction is usually the stronger backbone than read-time abstraction alone

The literature usually favors *write-time* abstraction—retrospective induction, ABSTRACT, or DISTILL during skill authoring—over read-time summarization alone. TRACE2SKILL (Ni et al., 2026)'s prevalence-weighted consolidation beats a read-time summarizer baseline; CASCADE's write-time distillation beats a retrieval-plus-rerank control; and SIMPLEMEM (Liu et al., 2026d) shows in Table 5 that removing write-time semantic compression drops LoCoMo average F1 from 43.24 to 31.29. XSKILL (Jiang et al., 2026a) extends this finding to multimodal agents: its Table 3 ablation shows that removing the Experience Manager drops VisualToolBench Average@4 by 4.09 points and removing the Skill Manager drops it by 3.62, while read-time task-decomposition/adaptation ablations are smaller. SKILLFLOW-BENCH supports file-level write-time abstraction but is not a clean write-time-versus-read-time ablation; SKILLGEN, SKILLOPT, and SKILLGRAD add direct evidence that skill quality improves when success/failure evidence is converted into a persistent artifact before deployment; and RAW-EXPERIENCE shows that extraction quality and consumption quality must be evaluated separately (Ma et al., 2026a; Yang et al., 2026b; Wang et al., 2026b; Huang et al., 2026c). K2-AGENT (Wu et al., 2026b)'s declarative-procedural split is architectural corroboration.

**Caveats.** The result is cleanest under reasonably stationary tasks. In non-stationary deployments and visually grounded multimodal settings, read-time adaptation layered on top of write-time abstraction can still be necessary. SKILLTTA (Wang et al., 2026d) is the boundary case: it synthesizes task-specific transient skills at read time, improving several benchmarks without maintaining a persistent library.

### 9.8 Cross-pattern observations

Three observations connect the patterns. Admission, verifier quality under RL, maintenance, and write-time abstraction are all forms of *write-time discipline*. Retrieval scaling and focused-library effects both argue that smaller can be better, through different mechanisms: retrieval resolution and distractor load. Finally, lifecycle benchmarks warn that skill *usage* is not skill *utility*; WILD-SKILLS shows this through retrieval realism, and SKILLFLOW-BENCH through high-use, low-gain settings. The open problems in §12 either exploit these patterns or ask why a method appears to violate one without visible cost.

## 10 Infrastructure

Dynamic-skill methods depend on packaging formats, storage backends, marketplaces, SDKs, and edit-execution pipelines. These choices were peripheral in 2023 but are load-bearing by 2026: they define the skill unit, determine scaling limits, shape cross-user aggregation, and decide which operators can be implemented at useful velocity.

Table 8 groups representative systems by structural constraints, not quality. The visible diagonals—flat storage with near-monotonic {ADD}, hierarchies with SPLIT-heavy maintenance, and DAG/ontology stores with COMPOSE-heavy workflows—explain why methods rarely transfer cleanly across infrastructure stacks: off-diagonal operators are often too expensive to run at the required edit velocity.

### 10.1 Packaging formats and the minimum viable unit

The surveyed literature uses four packaging formats. `SKILL.md` packages dominate the agent-skill regime: one directory per skill, with a descriptor, optional scripts, and an interface manifest. Public product/specification sources define the practical convention as a folder with `SKILL.md` metadata and progressive disclosure from name/description to full instructions and resources (Zhang et al., 2025; Anthropic, 2025). Function-calling schemas and MCP/plugin manifests dominate tool-as-skill systems, where the portable object is a typed

| Dim. | Class | Representative systems | Fast ops[a] | Costly ops[a] | Primary trade-off |
|---|---|---|---|---|---|
| Package | SKILL.md | SKILLCLAW, AUTOSKILL, METACLAW, MUSE-AUTOSKILL, SKILLDEX, public Agent Skills spec | ADD, REFINE | COMPOSE | portability/lock-in |
| | Tool schema | Function-calling + MCP/plugin manifests | ADD, COMPOSE | ABSTRACT | schema/capability |
| | Hybrid skill/memory | XSKILL, WEBXSKILL, MEMENTO, SAGER[b] | ADD, RERANK | MERGE, PRUNE | portability/ compat. |
| | Compiled corpus | CORPUS2SKILL, SKILLREPOMINING, SKILLFOUNDRY, SKILLSMITH | ADD, ABSTRACT | PRUNE | provenance/quality |
| | Trajectory | SIMPLEMEM, MUSE, SKILLDROID[b] | ADD, RERANK | ABSTRACT, DISTILL | read-time/write-time |
| Storage | Flat embedding | EVOSKILL, SKILLWEAVER, AGENTIC-PROPOSING | ADD | MERGE | admit/reorganise |
| | Hierarchical tree | AGENTSKILLOS, SKILLX, CORPUS2SKILL, UNI-SKILL | SPLIT | PRUNE | restructure/remove |
| | Graph / dependency | PSN, CODE-SHARP, GRAPH OF SKILLS, GRASP, WEBXSKILL, SSL, SKILLOPS | COMPOSE[c] | PRUNE | chain/remove |
| | Typed ontology | SKILLORCHESTRA | COMPOSE (typed) | REFINE | expressivity/cost |
| Market | Pull-model | Central curated registry; SKILLDEX, SKILLSVOTE | ADD, RERANK | MERGE | curation/scale |
| | Push-model | SKILLCLAW cross-user | ADD | MERGE, PRUNE | velocity/ provenance |
| | Hybrid | AUTOSKILL dual review | ADD, RERANK | MERGE, PRUNE | review/throughput |
| | Security scanner | SKILLSIEVE, MALICIOUS-OR-NOT, SEMANTIC-SUPPLY, CREDENTIAL LEAKAGE | PRUNE | REFINE | recall/false positives |
| Pipeline | Inline edit | PSN, CODE-SHARP | REFINE, REWRITE | PRUNE | velocity/rollback |
| | Log-and-apply | AUTOREFINE, AGENTSKILLOS, SKILLGRAD, SKILLOPS | PRUNE | REFINE | audit/latency |
| | Compile-and-serve | SKILLSMITH | ABSTRACT | REFINE | compactness/ fidelity |
| | Shadow exec. | METACLAW, SKILL0 | DISTILL | REFINE | safety/staleness |
| | Release audit | MEDSKILLAUDIT, AGENTDEVEL | PRUNE | REFINE | rigor/throughput |

[a] "Fast" / "costly" are relative within a class: listed entries are drawn from the ten-operator vocabulary (§7.1) and flag which operators a given infrastructure stack admits cheaply (unit-cost edits) versus which require a heavier mechanism (locking, re-indexing, cross-user consensus, or human review).

[b] SIMPLEMEM and MUSE are trajectory-memory systems in which episodes themselves play the skill-like role of retrievable artifacts; SKILLDROID compiles trajectories into executable GUI templates; XSKILL and MEMENTO are hybrid cases that pair skill documents with case / experience stores. We include these methods when their artifact is read by an agent loop as if it were a skill. The central dynamic-skills-vs-memory distinction is preserved by the *Artif.* and *Trig.* columns of Table 11.

[c] For PSN and CODE-SHARP, the graph is cheap for library-edit-level COMPOSE to *audit* (prerequisites are explicit); admitting a composed skill as a new library entry still requires a verifier pass, so the cheapness is structural, not free.

Table 8: **Four infrastructure dimensions (package, storage, market, pipeline) that materially constrain dynamic skill systems.** Each row classifies a representative deployment along the same categorical vocabulary so that the operator economy and the primary trade-off are readable at a glance. The "fast ops" / "costly ops" asymmetry is the core point of the table: infrastructure choices predetermine which operator vocabulary is economical, which is why the operator–storage diagonal of the master taxonomy (Table 11) is sparsely populated.

callable signature. Hybrid skill-memory stores, as in XSKILL, WEBXSKILL, and MUSE-AUTOSKILL (Jiang et al., 2026a; Wang et al., 2026g; Lin et al., 2026), pair a workflow with executable or experience evidence; MUSE-AUTOSKILL makes this explicit through per-skill memory. A fourth, less explicit format treats trajectories themselves as retrievable artifacts (SIMPLEMEM, MUSE) or compiles them into GUI templates (SKILLDROID) (Chen et al., 2026b). Recent infrastructure papers add package-manager, corpus-compiler, compiler-runtime, typed-contract, and structured-representation variants: SKILLDEX (Saha & Hemanth, 2026) gives skills scoped package semantics, CORPUS2SKILL (Sun et al., 2026) compiles an enterprise corpus into a navigable skill directory, SKILLSMITH (Xu et al., 2026) compiles skill packages into boundary-guided runtime interfaces, SKILLOPS (Song et al., 2026b) casts each executable skill as a contract with preconditions, operation, artifacts, validators, and failure modes, and SSL (Liang et al., 2026a) normalizes skill text into scheduling, structural, and logical fields for retrieval and risk review.

The formats trade portability against structure. `SKILL.md` travels across backbones but depends on the receiver's ability to follow prose; function schemas travel across models but assume the target tool exists; hybrid stores require both workflow and experience interfaces; trajectory memories grow freely but are task-specific. K2-AGENT's declarative-procedural pair and XSKILL's Markdown-plus-experience split are the closest attempts to bridge formats, but both still assume compatible tools and context interfaces.

## 10.2 Storage backends and scaling limits

Storage determines whether moderate-library-size retrieval degradation is conceded or delayed. Flat embedding indices (EVOSKILL, SKILLWEAVER, AGENTIC-PROPOSING) use dense retrieval over descriptions and are the class most directly implicated by the scaling evidence. Hierarchical stores (AGENTSKILLOS, SKILLX, CORPUS2SKILL, UNI-SKILL) push the degradation point outward; graph stores (CODE-SHARP, PSN, GRAPH OF SKILLS, GRASP, WEBXSKILL, SSL, SKILLOPS) expose skill–skill, page–skill, or representation-level relations to retrievers and audits; ontology stores (SKILLORCHESTRA) attach semantic categories for typed composition. SKILLOPS adds a maintenance-specific graph layer in which dependency, compatibility, redundancy, and alternative edges drive health diagnosis before downstream retrieval. SRA adds the benchmark view: a large skill corpus is not only a retrieval target but also an incorporation problem, because a correct top-$k$ set does not ensure need-aware loading (Su et al., 2026).

For dynamic systems, the key question is operator cost under the storage class. Flat indices make ADD cheap and MERGE expensive; hierarchies make SPLIT cheap and subtree PRUNE expensive; DAGs make COMPOSE easier to audit but PRUNE risky because descendants may depend on the removed node; ontologies make category inheritance cheap but category-changing REFINE expensive. This operator–storage coupling explains why the master table's off-diagonal combinations remain sparse.

## 10.3 Marketplaces, plugins, and cross-user aggregation

A 2026 development is the skill *marketplace*: a registry where skills authored by one user or team are listed, reviewed, ranked, and adopted by others. Marketplaces change admission from a single verifier score to an aggregate utility and moderation signal. AGENT SKILLS: DATA-DRIVEN ANALYSIS (Ling et al., 2026) measures the speed of the public Claude-style skill ecosystem; SKILLSVOTE (Liu et al., 2026c) adds governed collection, recommendation, skill-linked credit, and evidence-gated evolution; and AGENTSKILLS-WILD (Liu et al., 2026g), MALICIOUS-SKILLS-WILD (Liu et al., 2026f), MALICIOUS-OR-NOT (Holzbauer et al., 2026), SEMANTIC-SUPPLY (Saha et al., 2026), and CREDENTIAL LEAKAGE (Chen et al., 2026f) show why registry metadata, repository context, behavioral verification, semantic retrieval manipulation, and remediation workflows are now infrastructure, not a separate security appendix.

The literature distinguishes pull-model registries with central admission, push-model sharing as in SKILL-CLAW, hybrid registries with per-user overrides as in AUTOSKILL, governed evolution as in SKILLSVOTE, and package-manager models such as SKILLDEX with scoped installs and conformance checks. Pull models foreground retrieval at scale; push, hybrid, governed, and package-manager models foreground attribution, credit assignment, and governance.

## 10.4 The edit–execute pipeline

Each operator in §7.1 must be implemented as an edit to a concrete artifact. Three pipeline architectures recur. *Inline editing* (PSN, CODE-SHARP, CONTRACTSKILL) edits the artifact in place and needs an explicit version store for rollback. *Log-and-apply* pipelines (AUTOREFINE, AGENTSKILLOS, SKILLDROID, SKILLGRAD, SKILLOPS) stage or schedule the edit, verify it, and commit only after admission, successful replay, or a health-triggered maintenance pass. *Shadow-execution* pipelines (METACLAW, SKILL0) evolve a shadow library while serving from the main one, then cut over during a distillation window. A fourth, *compile-and-serve* pattern is emerging in SKILLSMITH: a larger skill package is compiled into a smaller boundary-guided runtime interface before repeated use.

These pipelines determine whether a proposed operator is cheap enough to use continuously or expensive enough to reserve for release gates.

## 11 Safety and Governance

The 2026 safety wave turns dynamic skills from a prompt-injection concern into a software-supply-chain problem. The corpus now includes attack taxonomies, registry-scale measurements, admission scanners, confirmed malicious-skill datasets, credential-leakage studies, harmful-skill benchmarks, request-conditioned invocation audits, repository-context audits, and skill-stealing attacks. Because skill artifacts combine natural language, code, configuration, sometimes learned models, and social provenance, admission control, provenance, sandboxing, and release governance must be lifecycle stages.

Table 9 summarizes the evidence. The field has attack studies and partial defenses, but few defended lifecycle systems: methods verify utility but not maliciousness, scanners analyze artifacts but not downstream composition, and registries expose popularity but not operator-level lineage. Dynamic skill stores need a security pipeline that mirrors their lifecycle pipeline.

### 11.1 Eight safety surfaces specific to dynamic skills

The surfaces below are specific to, or strongly amplified by, lifecycle-managed skill stores. TOWARDS SECURE AGENT SKILLS (Li et al., 2026f) organizes threats across creation, distribution, deployment, and execution; we map that phase structure onto the survey's operators and artifact-store framing.

**Prompt injection through admitted skills.** An admitted skill can carry adversarial instructions in documentation, examples, comments, or generated helper files. AGENTSKILLS-PI (Schmotz et al., 2025) establishes the basic attack channel for Claude-style skill folders: skill metadata routes the agent to a file whose body or referenced scripts may contain hidden instructions. CLAWSAFETY (Wei et al., 2026) quantifies the outcome-level risk: malicious skill files are the highest-ASR injection vector in its OpenClaw personal-agent benchmark, averaging 69.4% ASR across backbones. SKILL-INJECT (Schmotz et al., 2026) adds benchmark-scale evidence with 202 injection-task pairs and contextual ASR reaching roughly the 40–80% range depending on model and setting. SKILLJECT (Jia et al., 2026) shows that attackers can optimize the skill artifact itself, reporting 95.1% average ASR after trace-driven closed-loop refinement versus 10.9% for naive injections. SKILLATTACK (Duan et al., 2026) shows the complementary red-team view: benign-looking skills can become exploitable when an attacker refines prompts into attack paths, with injected adversarial skills reaching high ASR and real-world Hot100 skills still exploitable under some conditions.

**Supply-chain poisoning at publication and installation.** Skill packages can be poisoned before a user ever invokes them. AGENTSKILLS-WILD (Liu et al., 2026g) measures broad vulnerability prevalence in public registries, analyzing 31,132 skills and reporting that 26.1% contain at least one vulnerability pattern; executable-script skills are 2.12× more likely to contain vulnerabilities than instruction-only skills. MALICIOUS-SKILLS-WILD (Liu et al., 2026f) provides confirmed-malice evidence: from 98,380 public skills, the authors identify 157 behaviorally confirmed malicious skills with 632 labeled vulnerabilities and report 93.6% removal after responsible disclosure. SUPPLY-CHAIN-POISONING (Qu et al., 2026) constructs DDIPE attacks over 1,070 adversarial skills across 15 MITRE ATT&CK categories and reports bypass rates from

| Surface | Most exposed life-cycle point | Current evidence | Required primitive | Coverage state |
|---|---|---|---|---|
| Skill-embedded prompt injection | admission / invocation | AGENTSKILLS-PI: simple SKILL.md/script channel; CLAWSAFETY: skill files highest-ASR vector (69.4% avg.); SKILL-INJECT: 202 injection-task pairs; SKILLJECT: 95.1% automated ASR | adversarial SKILL.md sanitization + request-conditioned invocation guard | attack measured; weak defense |
| Supply-chain poisoning | publication / install | AGENTSKILLS-WILD: 31,132 skills, 26.1% vulnerable; MALICIOUS-SKILLS-WILD: 157 confirmed malicious, 93.6% removed; SUPPLY-CHAIN-POISONING: 1,070 adversarial skills, 11.6–33.5% bypass; SEMANTIC-SUPPLY: discovery/selection/governance manipulation; BADSKILL: 97.5–99.5% ASR | package manifest, model provenance, sandboxed install, takedown workflow | attack measured; partial scanners |
| Harmful but valid skills | deployment / policy gate | HARMFULSKILLBENCH: 4,858 harmful of 98,440 collected skills; installed skills increase harm scores | capability-policy release gate | benchmarked; immature governance |
| Credential / secret leakage | execution / stdout / files | CREDENTIAL LEAKAGE: 520 affected of 17,022 sampled skills; 76.3% cross-modal NL+code cases; 73.5% debug-log vector | cross-modal secret scanner + stdout containment | measured; remediation partial |
| Scanner false positives and context-sensitive invocation | admission / registry review / invocation | AGENTSKILLS-WILD: SkillScan 86.7% precision / 82.5% recall; SKILLSIEVE: 0.800 F1, low-cost triage; MALICIOUS-OR-NOT: repository context leaves 0.52% of scanner-flagged skill-repo pairs malicious; STARS: request-conditioned audit; USER-COMPREHENSION: incomplete capability disclosure in specs | repository-aware triage, provenance scoring, invocation-time risk model, disclosure rubric | partial defense + calibration |
| Skill theft / IP leakage | marketplace / black-box use | SKILLSTEALING: proprietary skills can leak semantically through few interactions | ownership metadata, rate limits, watermarking, access control | measured; no mature defense |
| Domain release readiness | pre-deployment review | MEDSKILLAUDIT: 57.3% below Limited Release; system-expert ICC 0.449 on 75 medical skills | domain-specific audit rubric and release tiers | early domain audit |
| Attribution / composition misuse | maintenance / composition | ASG-SI audited graphs; SKILLORCHESTRA typed routing; no full operator-level audit | operator-granular lineage + composition-time review | architectural only |

Table 9: **Safety surfaces for dynamic skill systems after the 2026 safety wave.** The table distinguishes attack evidence, defense evidence, and governance primitives. Skill safety is not a single prompt-injection benchmark: it includes supply-chain poisoning, harmful-but-valid capability packages, credential leakage, scanner calibration, invocation-time risk, skill theft, and domain release readiness.

11.6% to 33.5% depending on defense configuration, with 2.5% evading both static and alignment filters. SEMANTIC-SUPPLY (Saha et al., 2026) adds the registry-facing semantic channel: SKILL.md triggers and descriptions manipulate discovery, selection, and governance even when no executable exploit has run. BAD-SKILL (Tie et al., 2026) adds a distinct model-in-skill threat: a skill can bundle a backdoored model whose malicious behavior is hidden in learned parameters, reaching 97.5–99.5% ASR while largely preserving benign accuracy.

**Harmful but well-formed skills.** A skill can be valid, useful, and non-poisoned while still enabling harmful functionality. HARMFULSKILLBENCH (Jiang et al., 2026c) finds 4,858 harmful skills among 98,440 collected skills and shows that installed harmful skills can raise harm scores even under implicit invocation. This is a policy-governance problem rather than malware detection.

**Credential and secret leakage.** Credential leakage is a cross-modal property of skills, because secrets can be exposed only through the interaction of SKILL.md text, code, runtime stdout, and agent context. CREDENTIAL LEAKAGE (Chen et al., 2026f) samples 17,022 skills from a 170,226-skill SkillsMP population, identifies 520 affected skills and 1,708 issues, and reports that 76.3% of cases require joint NL+code analysis. Debug logging accounts for 73.5% of vulnerability issues because stdout is often fed back to the LLM, turning routine logs into credential disclosure.

**Admission scanner limits and false positives.** Security scanners are necessary but not sufficient. AGENTSKILLS-WILD reports a broad SkillScan detector with 86.7% precision and 82.5% recall, while SKILLSIEVE (Hou & Yang, 2026) is one of the strongest current defense-side triage results, filtering 86% of skills at zero API cost and reporting 0.800 F1 at about $0.006 per skill. MALICIOUS-OR-NOT (Holzbauer et al., 2026) shows the opposite failure mode: scanner-only classification can badly overstate risk, while repository context reduces 2,887 scanner-flagged skill-repository combinations to 0.52% remaining in malicious flagged repositories. STARS (Zhang et al., 2026c) adds the invocation-time view: the risk of calling a skill depends on the request and context, and its calibrated fusion model improves held-out indirect-prompt-injection HR-AUPRC over static-only and contextual-only scorers. USER-COMPREHENSION (Wen, 2026) adds a human-facing audit gap: many skill specifications do not expose enough operational basis, output contract, boundary disclosure, or examples for users to form bounded expectations. Admission gates therefore need content analysis, provenance/context analysis, request-conditioned invocation review, and user-facing capability disclosure.

**Ownership, confidentiality, and skill theft.** Once skills become valuable artifacts, the market creates confidentiality and intellectual-property risks. SKILLSTEALING (Wang et al., 2026h) studies black-box extraction from proprietary agents and reports that paid or proprietary skill behavior can be inferred with only a few interactions, with substantial semantic leakage even when exact text recovery is low. This surface is orthogonal to user harm: the victim may be a skill author or registry operator rather than the end user.

**Domain-specific release readiness.** High-stakes domains need release gates beyond generic maliciousness. MEDSKILLAUDIT (Hou et al., 2026) evaluates 75 medical research skills, finds that 57.3% fall below the Limited Release threshold, and reports automated-audit ICC(2,1)=0.449 against expert consensus. The result supports domain-specific rubrics for scientific integrity, reproducibility, and boundary safety.

**Attribution loss and composition misuse.** Maintenance operators can delete, merge, split, and rewrite skills. Without operator-level provenance, later audits cannot reconstruct which artifact contributed to a decision. Composition adds another risk: individually acceptable skills can be harmful when chained. ASG-SI (Huang & Huang, 2025) is the closest surveyed system to graph-level audit, and SKILLORCHESTRA (Wang et al., 2026c) gestures toward typed composition, but neither closes the full provenance-plus-composition safety loop.

### 11.2 Coverage gap across the surveyed literature

The table's main implication is that skill safety is a lifecycle property: the vulnerable point can be admission, publication, installation, execution, marketplace review, or later composition.

### 11.3 Governance primitives a deployment would need

A defensible deployment needs eight primitives: an *admission-time scanner* combining static checks, LLM/rubric analysis, sandboxing, and repository provenance; a *semantic registry monitor* for trigger/description manipulation in discovery and selection; a *package manifest* for dependencies, permissions, model artifacts, network endpoints, and allowed tools; a *request-conditioned invocation audit* for deciding whether a skill should be called in the current context; a *runtime containment layer* for stdout, filesystem, network, and bundled-model channels; an *operator-granular provenance log* for ADD, REFINE, MERGE, SPLIT, and PRUNE; a *policy and domain release gate*; and a *market governance layer* for ownership, takedown, remediation, abandoned-repository hijacking, and user-facing capability disclosure.

If a system can create, modify, distribute, retrieve, and compose skills automatically, it must also audit those operations automatically.

## 12 Open Problems

The seven evidence-graded patterns constrain the next research agenda: new methods should exploit them or explain why their setting violates them. This section states eight open problems, organized around admission, maintenance, retrieval/composition, and deployment. We label each problem's evidence basis as *measured* when it follows from direct benchmark or deployment evidence, *extrapolated* when it extends measured behavior beyond the tested regime, and *speculative* when the literature has not yet run the necessary horizon or scale. Table 10 gives a concrete first experiment for each.

### 12.1 Compositional verifiers for code skills

*Evidence basis: extrapolated from measured verifier ablations.* Verifier design is decisive in skill-aware RL, and PSN's rollback-gated behavioral validation and CODE-SHARP's learned judges expose a useful trade-off: probe-set behavioral coverage versus broader but less grounded judge coverage. No surveyed method composes them.

### 12.2 Admission under non-stationary task distributions

*Evidence basis: extrapolated from deployment and focus/maintenance evidence.* Admission and focus interact badly under non-stationary deployment: yesterday's verifier can become a poor predictor of today's utility, and yesterday's focused library can become harmful. Existing systems (SKILLCLAW, AUTOSKILL, META-CLAW) use eviction but do not adapt the verifier itself. The open problem is a verifier whose threshold and scoring function update from downstream utility.

### 12.3 Principled maintenance schedules

*Evidence basis: measured need, extrapolated scheduling model.* Maintenance becomes important once libraries grow, but published schedules are engineering defaults. AUTOREFINE runs per-episode maintenance, META-CLAW/SKILL0/K2-AGENT trigger distillation periodically or by prevalence, and AGENTSKILLOS audits on a human-scale cadence. The open problem is to derive when to invoke PRUNE, MERGE, or DISTILL from monitored quantities such as admission rate, drift rate, retrieval entropy, and maintenance cost. Online learning with restarts is one possible analogue.

| Open problem | Evidence basis | Concrete first experiment | Likely obstacle |
|---|---|---|---|
| Compositional verifiers for code skills. | extrapolated | Replace one CODE-SHARP mutation with a PSN-style rollback-gated variant; measure velocity-vs-admission frontier. | Different graph semantics (invocation vs prerequisite). |
| Admission under non-stationary task distributions. | extrapolated | Wrap a skill-aware RL verifier in a utility tracker that shifts the admission threshold; test on a planted distribution-shift benchmark. | No such benchmark currently exists. |
| Principled maintenance schedules. | measured + extrapolated | Derive optimal maintenance frequency $f^*$ from a cost model; compare to the periodic-distillation defaults used by two-timescale systems (METACLAW, SKILL0). | Bounds likely depend on unobserved drift rate. |
| Parametric-collapse prevention under repeated distillation. | speculative | Run any two-timescale system for $10\times$ its reported distillation windows; track proposal diversity. | Compute budget for long horizons. |
| Retrieval scaling beyond moderate flat-library sizes. | measured + extrapolated | Evaluate learned-routing, graph retrieval (GRAPH OF SKILLS), skill-compilation, and hybrid-parametric baselines at 10,000–1,000,000 skills (e.g., against AGENTSKILLOS's 200K capability-tree curve). | Apples-to-apples comparison across storage structures is not yet standardized. |
| Cross-library skill portability. | measured + extrapolated | Extend COEVOSKILLS / XSKILL-style transfer into a controlled cross-product of author backbone, receiving backbone, harness, context budget, and tool API. | Tool / tokenizer / context mismatches; transfer can increase exploration while lowering average rollout quality. |
| Governance, provenance, and attribution. | measured + extrapolated | Scale ASG-SI-style audited provenance plus SKILLSIEVE-style triage: log-based "logical undo" vs cryptographic provenance vs snapshot at $10^3$ edits/day. | Provenance and scanner overhead under high operator velocity. |
| Benchmarks honest about dynamics. | measured | Combine SKILLFLOW-BENCH's sequential patch protocol, WILD-SKILLS' retrieval realism, RAW-EXPERIENCE's extractor/consumer split, and SKILLSVOTE's skill-linked credit; report global library trajectory, operator velocity, and usage-vs-utility. | Standardizing a shared global-library protocol across task families. |

Table 10: **Eight open problems for the dynamic-skills research community, each paired with a concrete first experiment.** The first four rows are method-level problems tractable for a single team; the last four require infrastructure or community coordination. The "likely obstacle" column states the main technical blocker for planning a research program.

## 12.4 Preventing parametric-collapse under repeated distillation

*Evidence basis: speculative.* Two-timescale systems may risk a degenerative echo: each distillation window compresses skills into the model, and the model then proposes skills resembling what it just absorbed. No surveyed method (METACLAW, SKILL0, K2-AGENT) runs long enough to test this directly. The missing evidence is long-horizon behavior under repeated distillation, especially proposal diversity and dependence between new proposals and recently distilled skills.

## 12.5 Retrieval beyond moderate library sizes

*Evidence basis: measured at moderate scale, extrapolated to marketplace scale.* Hierarchy, DAG, and ontology storage push the 64–128-skill flat-retrieval drop rightward but do not prove general removal. The open problem is scaling composition to the 10,000–1,000,000-skill regime suggested by SKILLROUTER's 80K pool, SRA's 26K-skill corpus, AGENTSKILLOS's 200K-scale evaluation, SKILLNET, and SKILLDEX. GRAPH OF SKILLS (Liu et al., 2026a) improves dependency-aware retrieval up to 2,000 skills, but marketplace-scale evidence is missing. Three directions are ready for comparison: learned routing over skill IDs (SKILLROUTER), skill compilation via MERGE/DISTILL (SINGLE-AGENT-SKILLS), and hybrid parametric–retrieval systems that retrieve only when routing confidence is low.

## 12.6 Cross-library skill portability and naming

*Evidence basis: measured transfer cases, extrapolated compatibility model.* Skills increasingly move across agents and users (SKILLCLAW, AGENTSKILLOS, AUTOSKILL), but the literature treats portability as deployment rather than learning. The open problem is *capability compatibility*: when is a skill authored on agent $A$ safe and useful on agent $B$? Tool API, tokenizer, context window, and harness assumptions can all break transfer. COEVOSKILLS reports broad cross-model transfer, while XSKILL reports mixed multimodal transfer; a useful compatibility model should predict such outcomes before evaluation.

## 12.7 Governance, provenance, and attribution

*Evidence basis: measured attacks and partial defenses, extrapolated integration.* §11 shows fast growth in attack and measurement work but few integrated defended systems. The open problem is making ASG-SI-style audited graphs, evidence bundles, and verifier-gated promotion (Huang & Huang, 2025) compatible with $10^2$–$10^3$ fast-loop edits per day and marketplace-grade triage. AGENTSKILLS-WILD, MALICIOUS-SKILLS-WILD, SKILL-INJECT, SKILLJECT, SKILLSIEVE, STARS, CREDENTIAL LEAKAGE, and MALICIOUS-OR-NOT (Liu et al., 2026g;f; Schmotz et al., 2026; Jia et al., 2026; Hou & Yang, 2026; Zhang et al., 2026c; Chen et al., 2026f; Holzbauer et al., 2026) supply pieces of the admission and invocation scanner; missing are operator-level provenance and costed deployment architectures. Candidate mechanisms include logical undo over operator logs and cryptographic provenance linking each skill to the trajectory that justified admission.

## 12.8 Benchmarks that are honest about dynamics

*Evidence basis: measured benchmark gaps.* SKILLFLOW-BENCH is the clearest temporal benchmark for discovery, patching, reuse, and repair; WILD-SKILLS is the clearest benchmark for realistic retrieval and curation utility; SWE-SKILLS-BENCH is the clearest paired evidence that many public skills add no marginal utility in software engineering; RAW-EXPERIENCE separates extraction from consumption; SKILLSVOTE adds skill-linked credit assignment; and SRA is the clearest decomposition of retrieval, incorporation, and downstream use. The open problem is to combine temporal patching with retrieval realism while reporting global library trajectory, operator velocity, and usage-vs-utility.

Together, the first four problems improve algorithms inside existing mechanism families, while the last four require community infrastructure for portability, governance, and trajectory-aware benchmarks. Method papers and infrastructure papers should therefore be judged together: dynamic skills will not mature if algorithms improve inside isolated libraries while portability, provenance, and benchmark realism lag behind.

## 13    Limitations and Update Policy

This survey draws analytical conclusions about a young area, so its limitations matter. The corpus is frozen at the May 31, 2026 cutoff in §2.3; later work may change the evidence for individual patterns, especially in registry-scale retrieval and safety. The taxonomy is therefore an updatable frame, not a final vocabulary.

The search protocol is broad but not exhaustive. The corpus was assembled through iterative search and snowballing over a fast-moving arXiv-heavy area, so false negatives are likely, especially for unpublished systems, product documentation, non-English papers, and papers that use tool, memory, workflow, or curriculum terminology without the word "skill". The cutoff should therefore be read as an audit boundary over the cited literature, not as a claim that every adjacent artifact-learning paper has been enumerated.

The evidence is heterogeneous. Method papers, benchmarks, infrastructure papers, and safety studies do not support the same kinds of claims: a verifier ablation is stronger evidence for an algorithmic pattern than a single benchmark score, while a registry-scale measurement is stronger evidence for a deployment surface than for a remedy. We use the lifecycle framework to align evidence roles rather than rank all papers on one axis.

The definition of "skill" remains unstable across executable programs, natural-language lessons, graphs, adapters, memories, and capability labels. Our six-sense terminology and skill-record schema make the boundary explicit, but generic tool-use papers, memory-only agents, and fine-tuning pipelines are excluded unless they produce an externally invocable artifact or evaluate that artifact's lifecycle.

The lifecycle framing is also imperfect. It fits systems that store reusable artifacts outside the model, but purely parametric methods compress proposal, verification, and admission into a training loop; memory-only methods may retrieve episodes without ever forming an invocable interface; and multimodal or embodied systems may attach verification to a simulator, visual affordance, or physical rollout rather than to a textual artifact. We include such systems only when they expose enough artifact structure to compare lifecycle choices.

The operator vocabulary is intentionally coarse. It does not specify closure, associativity, or edit-composition laws. It also does not model every edit dependency, such as a REFINE that triggers downstream PRUNE, a graph rewrite that changes both retrieval and composition semantics, or a stochastic proposal distribution that emits many candidates before one is admitted. The vocabulary is useful for comparing operator repertoires and storage costs, but it is not a full operational semantics for agent development environments.

Most conclusions are architectural rather than causal. The evidence supports patterns such as "admission gates matter", "flat retrieval has a moderate-size failure mode", and "benchmarks under-report library trajectories", but not a single prescription for the best verifier, storage topology, maintenance schedule, or distillation cadence. We also do not provide a quantitative cross-paper leaderboard: backbone, context budget, tool surface, evaluator, and task stream vary too much for pooled effect sizes to be meaningful without a shared harness.

## 14    Broader Impact Statement

Dynamic skill systems can make LLM agents more useful by preserving verified procedures, reducing repeated trial-and-error, improving transfer across related tasks, and exposing reusable artifacts for human inspection. These benefits matter most in operational settings where agents repeatedly use the same tools, interfaces, and domain workflows.

The same mechanism also creates new risks. A skill library is a high-trust execution substrate: admitted artifacts can carry prompt-injection text, unsafe code, bundled model backdoors, stale procedures, hidden credential leaks, or harmful but well-formed capabilities. Registry and marketplace settings add supply-chain, ownership, attribution, and skill-stealing concerns. This survey does not release new agent capabilities, but it organizes design patterns that could be used to build more autonomous skill-generation pipelines. The risk is highest if readers adopt growth and sharing mechanisms without the corresponding admission, containment, provenance, and rollback mechanisms analyzed in §11.

The mitigation message of the survey is therefore structural. Dynamic-skill research should treat safety and governance as lifecycle stages rather than after-the-fact filters: candidate skills should pass admission-time checks, carry manifests for permissions and dependencies, execute inside containment boundaries, retain operator-level provenance, and support demotion or rollback when downstream evidence changes. Benchmark papers should report unsafe invocation, skill usage versus skill utility, and maintenance-off behavior when those quantities are relevant to deployment.

## 15 Conclusion

When agents store reusable procedural knowledge outside the model, the skill library becomes part of the learning system. It has state, update rules, selection pressure, memory compression, maintenance costs, and safety constraints. Four survey takeaways are now well supported; one scaling boundary remains open.

The first takeaway is *the lifecycle view*. Dynamic skill systems are lifecycle-managed, verified, evolving artifact stores. The decomposition in §5 makes SKILLWEAVER, PSN, SKILLROUTER, METACLAW, SKILLFOUNDRY, SKILLDEX, CLAWSAFETY, SKILLFLOW-BENCH, SKILLOPT, and SKILLSVOTE comparable without pretending that they solve the same problem.

The second takeaway is *the time index*. Viewing libraries as dynamic objects $\mathcal{L}_t$ makes maintenance-off ablations, library-size drops, verifier-quality sensitivity, usage-vs-utility gaps, and two-timescale decoupling expressible. The skill-record schema $\langle C, \pi, T, R, \varphi, \nu, \prec \rangle$ provides the structural language for those analyses.

The third takeaway is *write-time discipline*. Admission, verifier quality under RL, maintenance, and write-time abstraction all concern what enters the library, under what gate, and with what abstraction. The newest skill-optimization papers make the point explicit: the editable skill artifact is the trainable external state, and validation decides which textual updates survive. The evidence aligns with a broader lifelong-learning principle: what a system keeps can matter more than what it sees.

The fourth takeaway is *mechanism pluralism*. The operator vocabulary describes multiple coherent specializations: retrospective induction, execution-verified skill writing, skill-aware RL, cross-user aggregation, graph/hierarchy management, and two-timescale internalization. These are not separate taxonomies; they are different ways of coupling edit repertoire, admission, storage, and update clocks.

The open boundary is scaling. Moderate-library-size retrieval degradation is recurrent; hierarchy, DAG, ontology, routing, and orchestration push it rightward but have not shown general removal. SKILLFLOW-BENCH improves evaluation by tracking discovery, patching, repair, skill count, and usage, but its family-local protocol does not answer global heterogeneous-library scaling. Retrieval scaling, cross-library portability, provenance, and benchmark honesty remain open.

**Reporting checklist.** For future work to become comparable, papers should report five quantities:

1. which lifecycle stages are implemented and which are only assumed;

2. operator velocities and library trajectories over time, not only final task success;

3. repair-, maintenance-, or admission-off ablations when those stages are part of the method;

4. skill usage separately from skill utility;

5. the operators that an infrastructure or safety substrate makes cheap, expensive, blocked, or auditable.

Safety papers should additionally evaluate admission, composition, and provenance surfaces rather than treating skills as ordinary prompt files. If these quantities become standard, the next survey can make quantitative cross-method comparisons that this one deliberately avoids.

## Acknowledgments

This research was supported by the National Institute of Standards and Technology under Federal Award ID 60NANB24D231 and by Carnegie Mellon University's AI Measurement Science and Engineering Center (AIMSEC).

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

# A Coding Protocol and Audit Materials

This appendix records the audit machinery behind the taxonomy. Method comparisons in the main text are based on mechanisms and evidence roles rather than on recency.

## A.1 Screening and Coding Fields

Each note was coded for: problem framing, artifact type, update locus, trigger, operator repertoire, learning signal, storage topology, verification or admission mechanism, evaluation setting, headline results, ablations, limitations, closest peers, and survey takeaway. Papers were included in the primary method cluster if they changed a skill artifact or evaluated the lifecycle of such artifacts. Modern papers were treated as boundary/context within the broader audit set if they studied generic lifelong learning, generic tool use, memory-only agents, or adjacent self-evolution settings without an externally invocable skill artifact. Older options and hierarchical-RL papers were cited as background anchors, not coded as modern audit records.

The coding fields intentionally separate *what changes* from *how evidence supports the synthesis.* For example, a paper may be coded as an executable-library method because it edits code skills, while its evidence role may be benchmark, deployment measurement, or safety audit depending on what the paper actually measures. This separation is why the main text avoids cross-paper leaderboards unless the benchmark harness is shared.

## A.2 Evidence Grades

The evidence grades used in Table 7 are qualitative audit labels, not statistical confidence intervals. Grade A indicates multiple controlled ablations, or one clean ablation plus independent corroboration. Grade B indicates one controlled study, a strong benchmark protocol, or a deployment-scale measurement. Grade C indicates convergent benchmark behavior without a clean causal ablation. Grade D indicates architectural corroboration only. Mixed labels such as A/B or B/C are used when the supporting papers differ in strength across artifact families.

## A.3 Master Coding Sheet

Table 11 is the full representative coding sheet used to audit the lifecycle taxonomy. It is placed in the appendix because its role is reproducibility rather than narrative: the main text uses the family table and heatmap for synthesis, while this table gives reviewers a way to trace each representative system back to the seven coding fields.

Table 11: **Master coding table for representative dynamic-skill systems across the seven coding fields of Section 6, plus family cluster and one factual headline.** Cells use the legend below; the operator column uses single letters from the ten-element vocabulary of Equation 4. The table is a coding sheet that supports the lifecycle-family taxonomy rather than the primary conceptual taxonomy.
**Legend.** *Artifact:* Code = executable code, NL = natural-language heuristic, MD = SKILL.md, LoRA = parametric adapter, Mix = two or more. *Clock:* fast = in-context, slow = parametric, 2TS = two-timescale. *Trigger:* Task = task-end retrospective, Fail = failure-driven, Per = periodic maintenance, User = author/user edit, RL = inside RL loop. *Operators:* A=ADD, R=REFINE, M=MERGE, S=SPLIT, P=PRUNE, D=DISTILL, B=ABSTRACT, C=COMPOSE, W=REWRITE, K=RERANK. *Signal:* Rew = env reward, Exec = execution feedback, Crit = NL critique, Judge = verifier/judge, XUser = cross-user aggregation, Teach = teacher distillation. *Storage:* flat, tree, DAG, graph, subsp, ontol.

| Method | Cluster | Artif. | Clock | Trig. | Operators | Signal | Store | Headline |
|---|---|---|---|---|---|---|---|---|
| *Foundational* | | | | | | | | |
| VOYAGER (Wang et al., 2023) | Found. | Code | fast | Task | A | Rew+Exec | flat | 3.3× items; 15.3× tech-tree |
| LATM (Cai et al., 2023) | Found. | Code | fast | Task | A | Exec | flat | Tool-maker + tool-user beats few-shot |

*Table 11 (continued)*

| Method | Cluster | Artif. | Clock | Trig. | Operators | Signal | Store | Headline |
|---|---|---|---|---|---|---|---|---|
| *Skill-aware RL* | | | | | | | | |
| SAGE (Wang et al., 2025a) | RL | Code | 2TS | Task | A,R | Rew | flat | Skill-integrated RL on AppWorld |
| SKILLRL (Xia et al., 2026a) | RL | Code | slow | RL | A,D | Rew | tree | Hier. skill-conditioned RL |
| AGENTEVOLVER (Zhai et al., 2025) | RL | Mix | 2TS | RL | A,R,D | Rew+Crit | flat | Self-question / navigate / attribute |
| CODE-SHARP (Bornemann et al., 2026) | RL | Code | 2TS | Fail | A,R,M,W | Rew+Judge | DAG | 4 mutations + prereq DAG |
| AGENTIC-PROP. (Jiao et al., 2026) | RL | Mix | slow | RL | A,R,P,D,B,C | Teach+Judge | DAG | 91.6% AIME-25 (30B, 11K traj.) |
| COS-PLAY (Wu et al., 2026a) | RL | Mix | 2TS | RL | A,R,P,D | Rew | flat | Co-evolves decision + skill bank |
| SKILLMASTER (Yang et al., 2026a) | RL | Mix | slow | RL | A,R,K | Rew+Exec | flat | Counterfactual skill utility |
| SKILLFLOW-REC. (Zhang et al., 2026f) | RL | Mix | 2TS | RL | A,R,P,K | Rew | flat | Flow diagnostics for evolution |
| *Heuristic / lesson memory* | | | | | | | | |
| ERL (Allard et al., 2026) | Lesson | NL | fast | Task | A,R | Crit | flat | Task-end retrospective lessons |
| RETROAGENT (Zhang et al., 2026g) | Lesson | NL | fast | Fail | A,R | Crit+Judge | flat | Dual intrinsic feedback |
| MUSE (Yang et al., 2025a) | Lesson | NL | fast | Per | A,R,D | Crit | tree | Long-horizon productivity memory |
| EVOLVER (Wu et al., 2025) | Lesson | NL | fast | Task | A,R,W | Crit | flat | Strategic-principle evolution |
| MEMENTO (Zhou et al., 2026a) | Lesson | MD | fast | Task | A,R,K | Crit+Judge | tree | Case memory + SKILL.md rerank |
| XSKILL (Jiang et al., 2026a) | Lesson | Mix | fast | Task | A,R,M,P,B,K | Crit+Judge | flat | Visual dual skill/exp. store |
| SAGER (Tao et al., 2026) | Lesson | NL | fast | User | A,R,W | XUser | flat | Per-user policy skills for rec. |
| EMBODISKILL (Ju et al., 2026) | Lesson | NL | fast | Fail | R,W | Crit+Exec | flat | Skill-aware embodied reflection |
| SKILLTTA (Wang et al., 2026d) | Lesson | NL | fast | Task | B,D,K | Exec | flat | Test-time transient skills |
| *Executable skill libraries* | | | | | | | | |
| SKILLWEAVER (Zheng et al., 2025a) | ExecLib | Code | fast | Task | A,R,P | Judge+Exec | flat | Propose-practice-synth.-hone |
| ASI (Wang et al., 2025b) | ExecLib | Code | fast | Task | A,R,B,C | Exec | flat | WebArena +23.5 vs static |
| WEBXSKILL (Wang et al., 2026g) | ExecLib | Mix | fast | Task | A,R,D,B,K | Exec+Judge | graph | Grounded / guided web skills |
| CONTRACTSKILL (Lu et al., 2026b) | ExecLib | Mix | fast | Fail | A,R,B | Exec | flat | VWA GLM 28.1 vs 9.4 self skill |
| SKILLDROID (Chen et al., 2026b) | ExecLib | Code | fast | Fail | A,R,D,B,K | Exec | flat | 85.3%, 49% fewer LLM calls |
| AGENTFACTORY (Zhang et al., 2026i) | ExecLib | Code | fast | User | A,R,C | Judge | DAG | Subagents as evolvable skills |
| EVOSKILL (Alzubi et al., 2026) | ExecLib | Code | fast | Fail | A,R,P | Judge+Exec | flat | Failure-driven + Pareto admission |
| COEVOSKILLS (Zhang et al., 2026d) | ExecLib | MD | fast | Task | A,R,B | Judge | flat | Co-evolving verifier loop |
| PSN (Shi et al., 2026) | ExecLib | Code | fast | Fail | A,R,M,W,K,P | Crit+Judge | graph | 5 refactors + symbolic credit |
| SKILLCRAFT (Chen et al., 2026c) | ExecLib | Code | fast | Task | A,C,R | Exec+Judge | DAG | Verified compositional MCP |
| LIVE-SWE (Xia et al., 2025a) | ExecLib | Code | fast | Task | A,R | Exec | flat | Runtime scaffold synthesis |
| CASCADE (Huang et al., 2025) | Lesson | Mix | fast | Per | A,D,P | Teach | tree | Scientific skill consolidation |
| SKILLX (Wang et al., 2026a) | ExecLib | Code | fast | Per | A,S,B,K | Judge | tree | Automatic hierarchical library |
| TRACE2SKILL (Ni et al., 2026) | ExecLib | MD | fast | Task | A,M,D | Judge | flat | Prevalence-weighted consolidation |

*Table 11 (continued)*

| Method | Cluster | Artif. | Clock | Trig. | Operators | Signal | Store | Headline |
|---|---|---|---|---|---|---|---|---|
| SKILLCLAW (Ma et al., 2026b) | ExecLib | MD | fast | User | A,R,M,K | XUser | flat | Cross-user skill evolution |
| AUTOSKILL (Yang et al., 2026c) | ExecLib | MD | fast | User | A,R | XUser | flat | Training-free personalized bank |
| AUTOREFINE (Qiu et al., 2026) | ExecLib | MD | fast | Per | A,R,M | Crit+Judge | flat | Dual-form skills + maintenance |
| MEMSKILL (Zhang et al., 2026e) | ExecLib | MD | fast | Task | A,R,K | Crit | tree | Meta-memory skill banks |
| WILD-SKILLS (Liu et al., 2026h) | ExecLib | Code | fast | Per | A,P,K | Rew+Judge | flat | Utility under realistic retrieval |
| CUA-SKILL (Chen et al., 2026d) | ExecLib | Code | fast | Task | A,B | Exec | flat | Parameterized GUI skills |
| ABSTRAL (Song et al., 2026a) | Infra | MD | fast | Task | A,R,C | Crit | graph | Multi-agent design via SKILL.md |
| SKILLFOUNDRY (Shen et al., 2026) | ExecLib | Mix | fast | Per | A,R,M,P,B | Exec+Judge | tree | Scientific contracts + provenance |
| SKILLFORGE (Liu et al., 2026e) | ExecLib | MD | fast | Fail | A,R,W | Judge | flat | Failure diagnosis in cloud support |
| SKILLMOO (Gong et al., 2026) | ExecLib | MD | fast | Fail | R,P,K | Judge+Exec | flat | Pareto pass/cost optimization |
| BILEVEL-MCTS (Huang et al., 2026a) | ExecLib | MD | fast | Per | R,W | Judge | flat | MCTS over package structure |
| METASURFACE (Huang et al., 2026b) | ExecLib | Mix | fast | Fail | A,R | Exec | flat | Physics-verified scientific skills |
| EVOAGENT (Zhang et al., 2026a) | ExecLib | Mix | fast | User | A,R,K,C | XUser+Judge | tree | Multi-file skills + delegation |
| MACRO (Fan et al., 2026) | ExecLib | Code | 2TS | Task | A,C,D | Rew+Exec | flat | Discovers composite medical tools |
| UNI-SKILL (Xie et al., 2026) | ExecLib | Mix | fast | Task | A,R,D,B,C,K | Exec | tree | SkillFolder robotic expansion |
| SKILLGEN (Ma et al., 2026a) | ExecLib | MD | fast | Task | A,R,B | Exec+Judge | flat | Net-effect verified synthesis |
| HASP (Liu et al., 2026b) | ExecLib | Code | fast | Fail | A,R,C,D | Exec+Teach | flat | Program-function interventions |
| SKILLOPT (Yang et al., 2026b) | ExecLib | MD | fast | Fail | R,W | Exec+Judge | flat | Held-out text-space optimizer |
| MUSE-AUTOSKILL (Lin et al., 2026) | ExecLib | MD | fast | Task | A,R,K | Exec+Judge | flat | Skill-level memory + tests |
| SKILLGRAD (Wang et al., 2026b) | ExecLib | MD | fast | Fail | R,W | Exec+Judge | flat | Text gradients + momentum |
| *Parametric / training-time* | | | | | | | | |
| SELAUR (Zhang et al., 2026b) | Param | LoRA | slow | Fail | D | Rew(unc) | subsp | Uncertainty-reshaped fail-rewards |
| SKILL0 (Lu et al., 2026a) | Param | LoRA | slow | Per | D,B | Teach | subsp | Helpfulness-decaying curricula |
| LSE (Chen et al., 2026e) | Param | Mix | slow | Task | D,R | Rew | subsp | Trained prompt-context evolution |
| SKILLSCRAFTER (Wang et al., 2026f) | Param | LoRA | slow | Per | D,M,K | Rew | subsp | LoRA bank + semantic subspace |
| K2-AGENT (Wu et al., 2026b) | Param | Mix | 2TS | Task | A,R,D | Rew+Teach | tree | Decl.-proc. co-evolution |
| METACLAW (Xia et al., 2026b) | Param | Mix | 2TS | Per | A,R,D | Rew | flat | Fast-skill / slow-weight adapt. |
| CO-EVOLVING (Jung et al., 2025) | Param | Mix | slow | Fail | A,D,P | Rew+Crit | flat | Hard-negative failure training |
| AGENT0 (Xia et al., 2025b) | Param | Mix | 2TS | RL | A,D | Rew | flat | Zero-data curric.-executor co-ev |
| TOOL-R0 (Acikgoz et al., 2026) | Param | Mix | slow | RL | A,D | Rew | flat | Dual self-play for tool-use |
| SCALAR (Zabounidis et al., 2026) | Param | Mix | slow | RL | D,B | Teach | flat | Co-adapt difficulty + env |
| ARISE (Li et al., 2026e) | Param | LoRA | slow | RL | D,K | Rew | subsp | Swarm PPO + PSO actions |

*Table 11 (continued)*

| Method | Cluster | Artif. | Clock | Trig. | Operators | Signal | Store | Headline |
|---|---|---|---|---|---|---|---|---|
| EXIF (Yang et al., 2025b) | Param | LoRA | slow | RL | D,B | Rew+Teach | flat | Exploration-first skill data |
| *Infrastructure & governance* | | | | | | | | |
| AGENTSKILLOS (Li et al., 2026b) | Infra | MD | fast | Per | A,C,S,P,K | Judge | tree | Capability tree + DAG orch. |
| SKILLROUTER (Zheng et al., 2026) | Infra | Code | fast | Per | K | Judge | flat | Full-text retrieval 80K skills |
| SKVM (Chen et al., 2026a) | Infra | Code | fast | User | A,R,C | Exec | DAG | Skills as compilable code |
| SKILLNET (Liang et al., 2026b) | Infra | MD | fast | Per | A,K,P | Judge | graph | Ontology + relation graph |
| SKILLORCHESTRA (Wang et al., 2026c) | Infra | MD | fast | Per | K,C | Judge | ontol | Skill handbooks for routing |
| SKILLFLOW-2025 (Li et al., 2025) | Infra | MD | fast | Task | K | Judge | flat | Multi-stage community-skill retrieval |
| SKILLFLOW-BENCH (Zhang et al., 2026j) | Eval | MD | fast | Task | A,R,P | Crit+Judge | flat | 166 tasks / 20 task families |
| SWE-SKILLS-BENCH (Han et al., 2026) | Eval | MD | fast | Task | K | Exec | flat | 49 SWE skills; +1.2 avg. gain |
| SRA (Su et al., 2026) | Eval | MD | fast | Task | K | Exec+Judge | flat | 5.4K tasks / 26K-skill corpus |
| AUTOAGENT (Wang et al., 2026e) | Infra | Mix | 2TS | Task | A,R,C | Crit+Judge | DAG | Evolves tools, peers, and self |
| AGENTDEVEL (Zhang, 2026) | Infra | MD | fast | User | A,R,K,P | XUser | graph | Release engineering + lineage |
| GEA (Weng et al., 2026) | Infra | Mix | 2TS | Per | A,R,M | Judge+Rew | ontol | Group-based framework evolution |
| SINGLE-AG.SK. (Li, 2026) | Infra | Code | fast | Per | A,M,D,P | Judge | flat | Multi→single compilation |
| EFFISKILL (Wang et al., 2026i) | Infra | Code | fast | Per | A,B,K | Rew | flat | Mined operator skills |
| GRAPH OF SKILLS (Liu et al., 2026a) | Infra | Code | fast | User | K,C | Rew | graph | Dependency-aware retrieval |
| GRASP (Xia et al., 2026c) | Infra | Code | fast | Task | C,R | Exec | DAG | Typed DAG composition + repair |
| SKILLDEX (Saha & Hemanth, 2026) | Infra | MD | fast | User | A,K,P | Judge | tree | Package manager + scoped registry |
| CORPUS2SKILL (Sun et al., 2026) | Infra | MD | fast | User | A,S,K | Judge | tree | Navigable enterprise QA skills |
| SKILLREPOMINING (Bi et al., 2026) | Infra | MD | fast | User | A,B | Judge | flat | GitHub repo mining to SKILL.md |
| AGENTSKILLS-DATA (Ling et al., 2026) | Eval | MD | fast | Per | K,P | Judge | — | 40K+ public-skill ecosystem study |
| SSL (Liang et al., 2026a) | Infra | MD | fast | User | B,K | Judge | graph | MRR .649→.729; F1 .409→.509 |
| SKILLLEARNBENCH (Zhong et al., 2026) | Eval | MD | fast | Task | A,R | Judge | flat | Continual skill-generation benchmark |
| RAW-EXPERIENCE (Huang et al., 2026c) | Eval | MD | fast | Task | B,D | Exec+Judge | flat | Extraction-consumption lifecycle study |
| SKILLSVOTE (Liu et al., 2026c) | Infra | Mix | fast | Task | A,R,P,K | Exec+Judge | flat | Governed open-skill evolution |
| SKILLSMITH (Xu et al., 2026) | Infra | MD | fast | User | B,C,K | Exec | tree | Boundary-guided runtime interface |
| SKILLOPS (Song et al., 2026b) | Infra | Code | fast | Per | R,M,P,K,C | Exec+Judge | graph | Technical-debt maintenance layer |
| SKILLSINJECTOR (Li et al., 2026d) | Infra | MD | fast | Task | K,W | Exec | flat | Adaptive skill-context construction |
| *Safety & audit* | | | | | | | | |
| ASG-SI (Huang & Huang, 2025) | Safety | Mix | 2TS | Task | A,R,P,D | Rew+Judge | graph | Audited skill graph + verif. rewards |

*Table 11 (continued)*

| Method | Cluster | Artif. | Clock | Trig. | Operators | Signal | Store | Headline |
|---|---|---|---|---|---|---|---|---|
| CLAWSAFETY (Wei et al., 2026) | Safety | MD | fast | User | P | Judge | — | Highest-ASR vector in personal agents |
| SECURE-SKILLS (Li et al., 2026f) | Safety | MD | fast | User | P | Judge | — | Lifecycle threat taxonomy |
| AGENTSKILLS-PI (Schmotz et al., 2025) | Safety | MD | fast | User | P | Exec | — | Simple SKILL.md injection channel |
| SUPPLY-CHAIN (Qu et al., 2026) | Safety | MD | fast | User | P | Judge | — | DDIPE poisoning attacks |
| AGENTSKILLS-WILD (Liu et al., 2026g) | Safety | MD | fast | User | P | Judge | — | 31K skills; 26.1% vuln. |
| MALICIOUS-SKILLS-WILD (Liu et al., 2026f) | Safety | Mix | fast | User | P | Exec+Judge | — | 157 confirmed malicious skills |
| SKILL-INJECT (Schmotz et al., 2026) | Safety | Mix | fast | User | P | Judge | — | 202 injection-task pairs |
| SKILLJECT (Jia et al., 2026) | Safety | MD | fast | User | R,W,P | Exec+Judge | — | 95.1% automated ASR |
| SKILLSIEVE (Hou & Yang, 2026) | Safety | MD | fast | User | P | Judge | — | Hierarchical malicious-skill triage |
| SKILLATTACK (Duan et al., 2026) | Safety | MD | fast | User | P | Judge | — | Automated attack-path red teaming |
| BADSKILL (Tie et al., 2026) | Safety | Mix | fast | User | P | Exec | — | Model-in-skill backdoors |
| CREDLEAK (Chen et al., 2026f) | Safety | Mix | fast | User | P | Exec+Judge | — | Cross-modal secret leakage |
| HARMFULSKILLBENCH (Jiang et al., 2026c) | Safety | MD | fast | User | P | Judge | — | Harmful-but-valid skills |
| SKILLSTEALING (Wang et al., 2026h) | Safety | MD | fast | User | K | Judge | — | Black-box skill extraction |
| MALICIOUS-OR-NOT (Holzbauer et al., 2026) | Safety | MD | fast | User | P | Judge | — | Repository-aware classification |
| STARS (Zhang et al., 2026c) | Safety | MD | fast | User | K,P | Judge | — | Request-conditioned invocation audit |
| MEDSKILLAUDIT (Hou et al., 2026) | Safety | MD | fast | User | P,R | Judge | — | Medical release-readiness audit |
| SEMANTIC-SUPPLY (Saha et al., 2026) | Safety | MD | fast | User | K,P,W | Judge | — | SKILL.md semantic registry attacks |
| USER-COMPREHENSION (Wen, 2026) | Safety | MD | fast | User | R | Judge | — | Skill-spec disclosure audit |

