# OpenReview forum: "Dynamic Agent Skills: A Lifecycle Survey and Taxonomy of Evolving Skill Libraries"
_TMLR — Accepted by TMLR_

### Review · Reviewer_qUBJ · 2026-05-12

**Summary Of Contributions:**

The manuscript proposes a seven-tuple framework for LLM agent skills to provide a taxonomy of recent methodological trends. Unlike prior work which treats skills as static objects, it is argued that skills in the modern form should be understood as lifecycle-managed, verified, evolving artifact stores. Furthermore, the manuscript describes regularities such as curated skills outperforming unverified self-generated skills and describes open problems such as theoretically grounded maintenance schedules.

**Audience:**

Yes

**Audience Explanation:**

It is a survey that summarizes recent trends in LLM agents. I am convinced that this is interesting to a broader audience.

**Broader Impact Concerns:**

I do have broader impact concerns regarding LLM agent skills, but they are more regarding the methodoligical advances and not this survey article. I believe the manuscript does a sufficient job at raising awareness on ethical implications.

**Claims And Evidence:**

Yes

**Claims Explanation:**

It appears that all claims are supported.

**Requested Changes:**

### Title

I understand that such wimsical titles are popular these days. In this title, however, I am afraid that the humor appears a bit forced. In "They Are Not Static", it is not clear who "they" are. Instead, I suggest "Agents are Not Static: A Survey of Dynamic Agent Skills" or even better "A Survey of Dynamic Agent Skills".

### Abstract

I found it a bit hard to parse the abstract because there were some terms that I am not so familiar with. Most importantly, I was not familiar with the term "skill" before reading the abstract and while examples are listed for skills, they also did not help me too much. Would it be possible to add a high-level explanation for what "skill" means? I feel similarly about the term "externalize" that is used a few times. What does that mean and could it be exchanged for some simple word?

### Section 1.1

What are "SoK skills"? Please always clarify such acronyms.

### Section 3.2

- The definition for "skill" in equation (1) seems reasonable. However, I have a hard time understanding what this "reusable interface" object is. I suggest elaborating more on that.

### Section 3.3

- What is "Instr."? Please clarify.

- *"components are nontrivial"* What are trivial components? Please clarify.

### Figure 1

- This figure appears very beautiful and clear, on first sight. However, on second sight, one can see all the artifacts that come with AI-generated art: it is unclear what the robot in the "execution" box does and the "prune" box shows some sort of hybrid between scissors and a tree. This is a subjective point, but for scientific drawings I recommend simple and self-made drawings.

- Also, I cannot find any mention of the "audit log" that is in the visualization, in the main text. Did I miss it?

### Figure 2

Title and legend overlap. I suggest fixing this.

---

> ### Author Response · Authors · 2026-05-12
>
> Thank you very much for the constructive and encouraging review. We appreciate your positive assessment of the manuscript’s contribution and relevance to the TMLR audience.
>
> We are also grateful for the detailed suggestions regarding the title, abstract clarity, terminology, definitions, and figures. We agree that these points can improve the accessibility and presentation of the paper. In the revision, we will clarify terms such as “skill,” “externalize,” “SoK,” and “Instr.,” elaborate on the reusable interface definition, and revise the figures to address the noted issues, including the overlap in Figure 2 and the unclear artifacts in Figure 1.
>
> Once all reviews are released, we will carefully summarize the reviewers’ comments and revise the manuscript accordingly. Thank you again for the helpful feedback.

---

> ### Author Response · Authors · 2026-06-07
> **Post-Revision Response to Reviewer qUBJ**
>
> Thank you for the encouraging review and for the careful technical suggestions. We especially appreciate your active engagement in the discussion and your independent checking of some table entries. Our goal throughout the revision was to keep the information accurate while presenting it at an abstract enough level to help readers understand the structure of the field.
>
> We addressed the presentation issues you raised. The title is now more direct and professional: "Dynamic Agent Skills: A Lifecycle Survey and Taxonomy of Evolving Skill Libraries." The abstract now defines "skill" before using the term extensively, and we replaced or explained potentially unclear terms such as "externalize." We also expanded "SoK" on first use, clarified the reusable interface field, removed ambiguous shorthand such as "Instr.," and revised language around "nontrivial" components.
>
> We also updated the figures. Figure 2 was replaced with a simpler timeline to avoid overlap and make future corpus updates easier. For Figure 1, we agree that scientific figures should avoid ambiguity. We kept the figure because it communicates the lifecycle at a high level, but revised the surrounding text and caption so that the governance elements, including audit logs, provenance, and rollback, are explicitly grounded in the manuscript rather than appearing only as visual decoration.

---

### Review · Reviewer_jUSV · 2026-05-13

**Summary Of Contributions:**

This paper surveys dynamic or self-evolving skill systems for LLM agents. Its main claim is that agent skills should not be viewed as static prompt snippets or tool lists, but as lifecycle-managed, verified, evolving artifact stores. The paper proposes a seven-tuple formulation for skills, a library-level transition view, and a ten-operator vocabulary covering operations such as Add, Refine, Merge, Prune, Distill, Compose, and Rerank. It then uses this framework to organize a 94-paper corpus around the lifecycle of evidence acquisition, proposal, verification/admission, storage, retrieval/composition, maintenance, distillation/portability, and governance.

The main strength of the paper is its coherent framing of a fragmented and fast-moving area. The lifecycle view is useful, especially the emphasis on admission gates, maintenance, retrieval scaling, provenance, and safety. The paper is also helpful in distinguishing executable skills, natural-language lessons, SKILL.md packages, parametric skills, memory traces, and capability labels.

The main weakness is that some of the higher-level claims are stronger than the underlying cross-paper evidence can fully support. The corpus is broad and useful, but heterogeneous. The proposed “operator algebra” is also more of a descriptive taxonomy than a formal algebra unless additional structure is specified.

**Additional Comments:**

One question for the authors: Which of the proposed empirical regularities remain strongly supported if the analysis is restricted only to primary dynamic-skill systems, excluding boundary cases such as memory-only systems, registry-only retrieval systems, and parametric-adapter systems?

**Audience:**

Yes

**Audience Explanation:**

This paper should be of interest to researchers working on LLM agents, tool use, lifelong learning, skill libraries, agent memory, evaluation, and safety. The topic is timely because many agent systems now externalize reusable procedures into files, code, skill packages, or registries. The paper gives this emerging area a useful vocabulary and organizes many disconnected papers into a common lifecycle view.

Even readers who do not work directly on “skills” may find the paper useful, because the issues it discusses—verification, maintenance, retrieval scaling, provenance, and unsafe reusable artifacts—are central to long-running agent systems more broadly.

**Broader Impact Concerns:**

I do not have major additional broader impact concerns beyond those already discussed in the paper. The paper correctly identifies dynamic skill libraries as a potential high-trust execution substrate and discusses risks such as prompt injection, unsafe skills, credential leakage, supply-chain poisoning, and skill theft.

The main broader impact point I would emphasize is that readers should not adopt growth and sharing mechanisms without corresponding admission checks, containment, provenance, and rollback. The paper already makes this point, but it could be stated even more directly in the final recommendations.

**Claims And Evidence:**

Yes

**Claims Explanation:**

Most of the paper’s central claims are supported by the surveyed literature. In particular, the lifecycle framing, the distinction between static and dynamic skill libraries, and the importance of verification, maintenance, retrieval, and provenance are well motivated by the cited systems and benchmarks.

However, the support is stronger for the conceptual taxonomy than for some empirical regularities. Claims such as retrieval degradation at moderate library sizes, weaker-backbone gains, or focused libraries outperforming comprehensive ones are plausible and useful, but the underlying studies differ in models, tasks, tool surfaces, context budgets, and evaluation harnesses. The authors are aware of this limitation, but some regularities should be stated more carefully as observed patterns rather than general laws.

Overall, I think the evidence is sufficient for a survey paper, but the paper would benefit from clearer separation between strongly supported claims, suggestive patterns, and architectural hypotheses.

**Requested Changes:**

The survey would be more actionable if it proposed a short checklist for future papers, such as reporting library size over time, admitted vs. rejected skills, operator counts, usage-vs-utility, verifier failures, retrieval distractor load, maintenance cost, rollback events, and provenance/safety checks.

---

> ### Author Response · Authors · 2026-05-15
>
> Thank you for the thoughtful review and for the concrete suggestion about adding a reporting checklist. We agree that this would make the survey more actionable, and we plan to add a concise checklist covering library size over time, admitted/rejected skills, operator counts, usage versus utility, verifier failures, retrieval distractor load, maintenance cost, rollback events, and provenance/safety checks.
>
> Regarding your question: if we restrict the analysis to primary dynamic-skill systems and exclude boundary cases such as memory-only systems, registry-only retrieval systems, and parametric-adapter systems, the strongest regularities remain: admission gates matter, verifier quality is load-bearing in skill-aware RL, and maintenance/repair becomes important as libraries grow. The retrieval-scaling regularity remains moderately supported, especially by controlled library-size studies and dynamic-library systems, but the large-registry evidence becomes contextual rather than primary. The weaker-backbone, focused-library, and write-time-abstraction patterns become more suggestive: they are still useful design hypotheses, but their evidence is more heterogeneous and partly supported by adjacent systems. We will revise the manuscript to make this distinction clearer by separating strongly supported findings, suggestive cross-system patterns, and architectural hypotheses.
>
> Once all reviews are released, we will carefully summarize the reviewers’ comments and revise the manuscript accordingly. Thank you again for the helpful feedback.

---

> ### Author Response · Authors · 2026-06-07
> **Post-Revision Response to Reviewer jUSV**
>
> Thank you for the constructive review and for the concrete suggestion to make the survey more actionable. We added a reporting checklist in the conclusion and strengthened the evaluation discussion around trajectory-aware reporting. The checklist now asks future papers to report implemented lifecycle stages, operator velocities and library trajectories over time, repair/maintenance/admission ablations, skill usage separately from skill utility, and which operators the infrastructure or safety substrate makes cheap, expensive, blocked, or auditable.
>
> We also strengthened the broader-impact message in the direction you suggested. The revised broader-impact section now states directly that dynamic skill libraries are high-trust execution substrates and that growth or sharing mechanisms should not be adopted without admission checks, containment, provenance, and rollback.
>
> Regarding your question about which empirical regularities remain strongly supported when restricting attention to primary dynamic-skill systems: we added an explicit answer in the regularities section. Under that restriction, the strongest patterns are admission, verifier quality, and maintenance/repair. Retrieval scaling remains moderately supported because it has both a controlled size sweep and registry-scale corroboration, but it still lacks a shared benchmark across storage designs. Weaker-backbone gains, focused-library advantage, and write-time abstraction are retained as useful cross-system signals rather than settled laws.

---

### Review · Reviewer_YXDu · 2026-05-26

**Summary Of Contributions:**

The authors perform a survey of LLM-based skills that are dynamic in nature, as well as propose an options-based formalism and subsequent taxonomy for these dynamic skills.

**Audience:**

Yes

**Audience Explanation:**

This paper would be of interest for individuals interested in LLM skill discovery.

**Claims And Evidence:**

No

**Claims Explanation:**

The authors make two primary claims in this paper:

1) That this paper serves a survey of dynamic LLM-based skills.
2) That their proposed options-based formalism is warranted.

In terms of the first claim, I find that the claim is not supported by the evidence provided in the paper. In particular, almost the entire paper focuses on introducing the options-based formalism and subsequent taxonomy. Conversely, the survey aspect of the paper is more or less confined to Section 3.1. The authors claim to have surveyed almost 100 papers, but the reader is only given a superficial level of understanding of some of these papers, with most of the discussion in the paper focusing on how these papers fit into the proposed formalism and taxonomy.

In terms of the second claim, I find that the claim is not supported by the evidence provided in the paper. In particular, the paper lacks a critical aspect of this formalism: motivation. That is, why is it necessary to introduce this formalism in a survey paper? One could make a counterargument that the categorization of the skills into the six senses described in Section 3.1 is sufficient for a survey paper, and the subsequent formalism and taxonomy is better served as a separate, follow-up paper.

**Requested Changes:**

My primary concern with this paper is that it positions itself as a survey paper but upon reading it, the survey aspect is relegated to the background and the focus is placed on the proposed formalism and taxonomy. In general, the aim of a survey paper should be to gently introduce the reader into the topic of interest by introducing key terminology, providing historical notes, etc. However, the survey aspects of this paper feel rushed and the paper assumes that the reader has a level of understanding that is much higher than what it would typically be for someone reading a survey paper.

Moreover, it is not clear to me why the formalism and most of the taxonomy is needed? One could make a counterargument that the categorization of the skills into the six senses described in Section 3.1 is sufficient for a survey paper, and the subsequent formalism and taxonomy is better served as a separate, follow-up paper.

I am also under the impression that the paper tries to introduce too much terminology. The authors begin by partitioning the papers into the “six senses”, then proceed to propose their options-based formalism, then introduce other types of partitions that will likely overload the reader. I challenge the authors to stick with one type of partition to minimize confusion.

In terms of the proposed options-based formalism, it is not clear whether the 3 additions to the skill tuple are best served as part of individual skills or if they would be better served as properties of the library itself. For example, can the authors think of an example from the survey papers where the set of skills within a given library would have different {edit operators, verification predicatives, lineage relations}? I would imagine in most cases the skills in a given library would share these operators, but I’m curious to see how this intuition lines up with the surveyed papers.

Finally, from a writing perspective, the paper is quite difficult to follow. The abstract and introduction contain large amounts of jargon that is not formally introduced until Section 3, which makes them difficult to follow. I challenge the authors to do a full re-write of the abstract and introduction and put them in "layman's terms”. Similarly, Section 2 would be better served as an appendix as it uses lots of terminology that is not introduced until Section 3. Finally, the organization of text as a whole feels incoherent; the need for Sections 4-9 is not clear. The writing is also, in my view, a bit too informal for an academic paper. For example, phrases such as “Flat retrieval exhibits a knee in the moderate-library-size regime, roughly one hundred skills” feel out of place in an academic paper.

---

> ### Comment · Reviewer_qUBJ · 2026-05-27
>
> First of all, I thank the reviewer for assessing the manuscript critically. However, I disagree with the statement that claims are not supported.
>
> (1) It is clear that the introduced formalism is used as a tool to organize the survey. That is, from my understanding, the very purpose of the formalism. I do not know all existing works in this domain (agent skills), but it is clear that the manuscript covers a broad range of recent works. Given that there exist so many works in this domain, I believe it is OK to remain "superficial" (I would rather call it "abstract") and focus on how to structure these recent trends well, which I believe the manuscript does well.
>
> (2) I also disagree that the formalism is not well-justified. There exist many tables that clearly show how existing methods differ in terms of the added components. For instance, table 2 provides clear examples of how different methods differ in terms of update rule operator and I checked for some of these methods whether the categorization seams plausible (which it does).

---

> > ### Comment · Reviewer_YXDu · 2026-05-27
> >
> > I thank the reviewer for their active engagement. I should clarify that my primary concern is not that the work presented is incorrect or without merit, but merely that it falls short as a survey paper. That is, because the authors themselves characterize the above work as a survey, then I must assess whether the work presented meets the bar of a survey paper. To that end, the paper falls short as the survey aspect is relegated to the background. For instance, consider a similar survey mentioned by the authors [1]. The authors of [1] are able to provide intuitive understanding of the various methods (for example, see Section 3.1 of [1] for the rigorous explanation presented for SKILL.md, and contrast that to the explanation presented in this work), all without needing to introduce a whole new formalism and framework.
> >
> > Now, it is possible that the formalism presented by the authors is needed, such that the survey aspect could not be performed without it. However, the onus is on the authors to rigorously justify the need of introducing the formalism in a survey paper. Based on the current draft of the paper, I am not convinced that the formalism is needed, and rather, I am under the impression that it obscures the survey aspect of the paper itself.
> >
> > As such, the authors need to be more rigorous if they wish to claim this work as a survey paper, or, alternatively, change the narrative of the paper to clarify that it is not a survey paper and that the purpose of the paper is to introduce a new formalism for dynamic LLM-based skills.
> >
> > I again thank the reviewer for their active engagement.
> >
> > [1] Renjun Xu and Yang Yan. Agent Skills for Large Language Models: Architecture, Acquisition, Security, and the Path Forward. arXiv:2602.12430, 2026

---

> > > ### Author Response · Authors · 2026-05-27
> > >
> > > We thank both reviewers for the thoughtful exchange. We especially appreciate Reviewer qUBJ’s careful engagement with the manuscript, including checking the table contents against some of the cited methods. This is very much aligned with our goal: to represent the surveyed works accurately while presenting the rapidly growing literature at an abstract level that helps readers see the structure of the field.
> > >
> > > We also understand Reviewer YXDu’s concern that this intent should be clearer for readers approaching the paper as a survey. Our goal is not to introduce formalism for its own sake, but to use it to make the comparison clearer: the formal lens abstracts over heterogeneous artifacts, update mechanisms, verification/admission procedures, and provenance so that different dynamic skill systems can be discussed in a common language. We will clarify this motivation and the relationship to Xu and Yan (2026) in the revision.
> > >
> > > Thank you again to both reviewers for the active engagement. Now that all reviewer comments have been posted, we will carefully consider the full set of feedback and begin revising the manuscript accordingly.

---

> > > > ### Comment · Reviewer_qUBJ · 2026-05-28
> > > >
> > > > I appreciate the engagement of Reviewer YXDu and the Authors.
> > > >
> > > > If Reviewer YXDu's point (1) ultimately boils down to the term “survey” being inappropriate, I would also be fine with the Authors simply not labeling the manuscript as a survey article and instead centering the work more explicitly around the taxonomy, for example in the title. This would also seem to address point (2), since, according to Reviewer YXDu, the lack of justification for the introduced formalism stems from the article being framed as a survey.

---

> ### Author Response · Authors · 2026-05-27
>
> Thank you for the careful and constructive review. We agree that the manuscript should make the survey framing and the motivation for the formal lens clearer to readers who are new to this area. Our intent is not to present the formalism as a separate theory, but to use it as a compact organizing language for comparing dynamic skill systems across artifact types, lifecycle stages, verification/admission mechanisms, maintenance, and provenance.
>
> We also appreciate the point about whether edit rules, verification predicates, and lineage are skill-level or library-level properties. Our intended view is that many systems share a library-level update policy, while individual skill records carry artifact-specific editability, verification handles, and provenance/lineage metadata. We will make this distinction explicit and reduce unnecessary jargon in the abstract and introduction. Thank you again for the helpful feedback.

---

> ### Author Response · Authors · 2026-06-07
> **Post-Revision Response to Reviewer YXDu**
>
> Thank you for the critical and detailed review. We agree that the original draft could feel too formal too early and could assume more prior familiarity with the agent-skill literature than a survey reader should need. We revised the title, abstract, and introduction accordingly. The revised paper now begins with a plain-language definition: an agent skill is a reusable procedure an agent can call later, such as a function, natural-language instruction, SKILL.md directory, workflow graph, or learned adapter. The six-sense taxonomy is now the first table in the paper and serves as the entry point before any formal notation is introduced.
>
> We also revised the manuscript to make the role of the formalism more modest and better motivated. The notation is no longer presented as a standalone theory. It is a lightweight skill-record schema and library-transition notation used to organize the survey: papers call very different artifacts "skills," and the schema makes it explicit whether a method exposes an editable body, reusable interface, verification handle, lineage, and library update. We added a worked AutoRefine-style example to show how the notation describes an actual library transition rather than merely naming components.
>
> On the question of whether the added fields belong to individual skills or to the library, we clarified the distinction in the formalism section. In most systems, the update policy is library-level; individual skill records expose the handles that the policy can use, such as an editable body, verification evidence, interface, and lineage. Cross-skill operations such as merge, split, compose, prune, and rerank are explicitly treated as library-level transitions.
>
> We respectfully keep the manuscript framed as a survey, because the central contribution is a comparative synthesis of a broad and fast-moving literature rather than a method proposal. At the same time, we now state more clearly that it is a taxonomy-driven survey, not an encyclopedic paper-by-paper tutorial. We also kept the corpus/scope protocol in the main text rather than moving it to the appendix, because the reviews asked for clearer survey motivation, scope, and evidence boundaries; putting the protocol in the main text makes those boundaries more auditable.

---

> > ### Comment · Reviewer_YXDu · 2026-06-16
> >
> > I thank the authors for their response as well as for the changes made to the paper. Although the updated paper is easier to follow in comparison to the earlier draft, it still does not satisfy my primary concern: the paper falls short as a survey paper. I emphasize to the authors that survey papers play a disproportionally important role and must therefore be of the highest quality. Currently, the paper focuses too much effort and space on explaining the proposed library-centric framework rather than on creating a readable synthesis of the field. Ultimately, upon reading it, I do not find that I have a better understanding of the field. More concretely, my concern is perhaps best shown with this sentence of the paper: "The central thesis of this survey is that dynamic skill systems are lifecycle-managed, verified, evolving artifact stores for LLM agents"; a survey paper should not have a thesis, its role is to create a readable synthesis of the field (e.g. see https://pmc.ncbi.nlm.nih.gov/articles/PMC4548566/). My recommendation to the authors is that if they insist on presenting the novel frameworks and formulations, they should do so under the pretext of a novel contribution, rather than as a survey paper.

---

> > > ### Author Response · Authors · 2026-06-18
> > >
> > > Thank you for the continued engagement and for clarifying the remaining concern. We understand the concern to be about genre and positioning: the paper should read as a synthesis of the field rather than as a framework paper that uses the literature mainly as support. We agree that survey papers have a high bar and should help readers understand the field, not merely introduce new terminology.
> > >
> > > We have made one additional wording change in direct response to the sentence you identified. The introduction no longer says "the central thesis of this survey." It now says: "This survey synthesizes the field through a lifecycle view: dynamic skill systems are lifecycle-managed, verified, evolving artifact stores for LLM agents." This better reflects our intent. The lifecycle view is an organizing lens for synthesizing the literature, not a new algorithm or standalone theoretical claim.
> > > We respectfully maintain that the manuscript is appropriately framed as a survey. Its primary object is the literature: it clarifies the overloaded meaning of "skill," separates artifact families, compares lifecycle stages implemented by existing systems, audits benchmarks and evidence strength, reviews safety and governance surfaces, and identifies open problems. The framework is included because the field is currently fragmented across executable skills, SKILL.md packages, natural-language lessons, graph systems, parametric skills, registries, and safety papers; without a common structure, the surveyed works are difficult to compare.
> > >
> > > That said, we take your readability concern seriously. In the revision, we moved the six-sense taxonomy to the introduction, reduced formal claims by replacing "operator algebra" with "operator vocabulary," added a concrete worked example, recalibrated empirical patterns as evidence-graded rather than universal, and clarified that the paper is a taxonomy-driven lifecycle survey rather than an encyclopedic paper-by-paper catalog.

---

### Review · Reviewer_foXH · 2026-05-29

**Summary Of Contributions:**

This paper surveys literature on dynamic agent skill systems for LLM-based agents, covering 94 papers from 2023 to 2026. It argues that agent skill libraries should be treated as dynamic, lifecycle-managed systems rather than static collections. The authors extend an options-based skill formalism into a seven-tuple, introduce a ten-operator vocabulary, and organize the literature around an eight-stage lifecycle architecture. Seven empirical regularities, a safety surface analysis, and a set of open problems are also included.

The main strength in my opinion is the six-sense taxonomy that distinguishes structurally different artifacts all called "skills" in the surveyed literature, and the lifecycle framing is also useful as it surfaces design decisions that individual papers tend to treat as implementation details.

The key weakness is with the overextension of formal terminology and the overly generous grading of empirical regularities.

**Audience:**

Yes

**Audience Explanation:**

The topic studied in this paper is timely and very relevant to TMLR's audience. Questions about how to verify, maintain, and govern growing skill libraries are becoming practically more and more important as agents are deployed in longer-horizon settings.

**Broader Impact Concerns:**

None. The discussion presented in the paper is adequate and I have no further concerns.

**Claims And Evidence:**

Yes

**Claims Explanation:**

The core argument is well-supported. The distinction between static and dynamic libraries is motivated throughout by concrete systems and benchmarks, and the lifecycle framing holds up across artifact types surveyed.

My concerns are mostly with part of the formalism and the empirical regularities.
1. The ten operators are called an "algebra" throughout the paper, but it is also noted that the set does not model edit dependencies, which means it lacks closure under composition. The operators are useful as controlled vocabulary for comparing systems, but the terminology seems an overstatement and should be corrected.

2. On the regularities, the evidence grades in Table 7 are somewhat generous. R1 and R2 are graded A, but the supporting ablations come from systems with different backbones, task surfaces, and evaluators, and it is not clear that aggregating them meets the bar implied by Grade A. Also, as defined by the authors' own methodology, Grade B includes findings backed by "one controlled study" and Grade C implies "convergent benchmark behavior without a clean causal ablation", which kind of undermines their labels as "empirical regularities".

**Requested Changes:**

1. Replace the word "algebra" with "vocabulary" or "taxonomy". The paper's own text already undermines the claim.

2. It may be better to revise the evidence grades in Table 7 to more honestly reflect the heterogeneity of the evidence. Revisit R1 and R2, particularly. A clearer distinction between strongly supported findings and suggestive cross-system patterns would make the regularities more credible.

Beyond these, I'd suggest moving Table 1 earlier, as it's the conceptual anchor for the whole paper and currently appears too late.

---

> ### Author Response · Authors · 2026-05-29
>
> Thank you for the constructive review. We are glad that the reviewer finds the core argument, six-sense taxonomy, and lifecycle framing useful.
>
> We agree that the term “algebra” overstates the role of the ten operators. Our intent is to provide a controlled vocabulary for comparing library updates, not a formal algebra with closure or compositional laws. We will replace this terminology with “operator vocabulary” or “operator taxonomy” throughout the manuscript.
>
> We also agree that the evidence grades should more clearly reflect cross-paper heterogeneity. We will revisit Table 7, especially R1 and R2, and distinguish more explicitly between controlled within-paper findings and suggestive cross-system patterns. Finally, we will move Table 1 earlier so that the six-sense taxonomy can serve as a clearer conceptual anchor for readers.

---

> ### Author Response · Authors · 2026-06-07
> **Post-Revision Response to Reviewer foXH**
>
> Thank you again for the helpful assessment. We agree with your two main concerns: the previous manuscript overstated the formality of the operator set, and some empirical grades were too generous given the heterogeneity of the evidence.
>
> In the revised manuscript, we replaced "operator algebra" with "operator vocabulary" or "operator taxonomy" throughout. We also added an explicit limitation stating that the vocabulary does not specify closure, associativity, edit-composition laws, or a full operational semantics. Its role is now narrower and, we think, more accurate: it gives a controlled language for comparing which library updates a system supports.
>
> We also revised the evidence section. The seven regularities are now presented as evidence-graded patterns rather than universal laws. In particular, R1 is graded A/B rather than A, R2 is graded B, the weaker-backbone pattern is graded C, and the table caption now emphasizes that grades are conservative qualitative audit labels rather than pooled effect sizes. This revision directly addresses the concern that the earlier labels could imply more cross-paper comparability than the literature supports.
>
> Finally, we moved Table 1 into the introduction. The six-sense taxonomy now appears before the lifecycle and update notation, so readers first see what current papers mean by "skill" before encountering the rest of the framework.

---

### Author Response · Authors · 2026-06-07

We thank all reviewers for the careful and constructive discussion. We have uploaded a substantially revised manuscript. The revision keeps the paper as a survey, but makes the framing more precise: it is now explicitly a taxonomy-driven lifecycle survey rather than an encyclopedic paper-by-paper catalog. We revised the title, abstract, and introduction to define agent skills in plain language before introducing notation, moved the six-sense skill taxonomy to the introduction as the conceptual entry point, and clarified how our scope differs from recent broader surveys of agent skills and self-evolving agents.

We also addressed the main technical concerns raised across the reviews. We replaced the term "operator algebra" with "operator vocabulary" or "operator taxonomy" throughout, and now state explicitly that the operators do not define closure, associativity, or a full operational semantics. We recalibrated the empirical regularities as evidence-graded patterns rather than universal laws, with more conservative grades and clearer caveats about heterogeneous backbones, tasks, context budgets, tool surfaces, and evaluators. We added a worked instantiation of the skill-record and library-transition notation, clarified the relation between skill-level records and library-level update/admission policies, strengthened limitations, broadened the reporting checklist, and made the safety/governance message more direct.

Because the area moved quickly during review, we also updated the audit set through May 31, 2026. The revised manuscript now covers 124 modern papers, with the cutoff and year counts defined in one place and reflected in the text, tables, bibliography, and timeline figure.

---

### Author Response · Authors · 2026-07-02
**Camera-Ready Version Submitted**

We thank the Action Editor for the recommendation of acceptance with Survey Certification, and all three reviewers for the careful reviews and active discussion throughout the process. The exchanges on framing, formal terminology, evidence grading, and accessibility substantially improved the paper.

We have submitted the camera-ready version with the requested minor editorial revisions. Specifically, we performed a final consistency pass to ensure the paper is framed throughout as a taxonomy-driven lifecycle survey rather than a new formal theory, verified that no residual "operator algebra" or universal-law language remains, aligned remaining wording with the latest revisions (e.g., open-problem labels now consistently use "principled" rather than "theoretically grounded" in both text and tables), and proofread the full manuscript for readability, consistent spelling, and minor presentation issues. No changes were made to the survey protocol, corpus, or structure.

Thank you again for the constructive and collegial review process.

---

> ### Comment · Action_Editor_B3RG · 2026-07-03
>
> Thank you for the camera-ready and the summary of changes. I checked the final version against the requested minor editorial revisions, and it appears to address them: the paper is consistently framed as a taxonomy-driven lifecycle survey, the former “operator algebra” language has been removed, the evidence claims are stated as evidence-graded patterns with caveats, and the open-problem terminology is consistent. I do not see any remaining issue requiring further revision.

---

### Decision · Action_Editor_B3RG · 2026-06-23

**Recommendation:** Accept with minor revision

**Additional Comments:**

I recommend acceptance with minor revision. The remaining required changes are minor and editorial: the authors should do a final consistency pass to ensure that the paper is framed throughout as a taxonomy-driven survey rather than as a new formal theory, remove any remaining language that suggests a formal algebra or unsupported universal laws, ensure that the most recent wording changes are reflected consistently, and proofread the final version for readability and minor presentation issues. No new experiments, new survey protocol, or major restructuring is required.

I also recommend Survey Certification. The paper provides a broad and useful survey/taxonomy of a fast-moving area, with a substantial audit set, a clear lifecycle organization, explicit evidence grading, safety/governance discussion, and actionable reporting recommendations for future work.

**Audience:**

Yes

**Audience Explanation:**

The topic is timely for researchers working on LLM agents, tool and skill use, lifelong learning, memory, retrieval/composition, evaluation, and agent safety. The paper is also relevant beyond the narrow "skill" terminology, because verification, maintenance, provenance, retrieval scaling, and unsafe reusable artifacts are central issues for long-running agent systems. The lifecycle taxonomy and reporting checklist are likely to be useful to researchers trying to compare or evaluate dynamic agent-skill systems.

**Claims And Evidence:**

Yes

**Claims Explanation:**

The submission's claims are supported by sufficiently accurate, convincing, and clear evidence for a taxonomy-driven survey. The central claims are synthetic rather than experimental: the paper distinguishes different senses of "skill", organizes a rapidly growing literature around lifecycle stages, introduces a lightweight vocabulary for library updates, and summarizes evidence-graded patterns and open problems. The initial reviews raised valid concerns about over-formalization, the term "operator algebra", the strength of some empirical regularities, and accessibility for readers. In the revised version, the authors substantially addressed these concerns: the "algebra" language was replaced by "operator vocabulary/taxonomy", the formalism is now framed as an organizing lens rather than a standalone theory, the empirical regularities are stated more conservatively, the abstract/introduction define skills more clearly, and the relationship to prior surveys is clarified.

One reviewer remains concerned that the paper is too framework-driven to qualify as a survey. I agree that this is not an encyclopedic tutorial survey. However, under TMLR's criteria, I do not view this as a fatal issue: the paper is explicitly positioned as a taxonomy-driven lifecycle survey, and the claims it makes in that capacity are adequately supported.